# Unbiased and efficient log-likelihood estimation with inverse binomial sampling

**Bas van Opheusden**[1,2☯*], **Luigi Acerbi**[1,3,4☯*], **Wei Ji Ma**[1,5]

**1** Center for Neural Science, New York University, New York, New York, United States of America,
**2** Department of Psychology, Princeton University, Princeton, New Jersey, United States of America,
**3** Department of Computer Science, University of Helsinki, Helsinki, Finland, **4** Department of Basic Neuroscience, University of Geneva, Geneva, Switzerland, **5** Department of Psychology, New York University, New York, New York, United States of America

☯ These authors contributed equally to this work.
* svo@princeton.edu (BVO); luigi.acerbi@helsinki.fi (LA)

## Abstract

The fate of scientific hypotheses often relies on the ability of a computational model to explain the data, quantified in modern statistical approaches by the likelihood function. The log-likelihood is the key element for parameter estimation and model evaluation. However, the log-likelihood of complex models in fields such as computational biology and neuroscience is often intractable to compute analytically or numerically. In those cases, researchers can often only estimate the log-likelihood by comparing observed data with synthetic observations generated by model simulations. Standard techniques to approximate the likelihood via simulation either use summary statistics of the data or are at risk of producing substantial biases in the estimate. Here, we explore another method, inverse binomial sampling (IBS), which can estimate the log-likelihood of an entire data set efficiently and without bias. For each observation, IBS draws samples from the simulator model until one matches the observation. The log-likelihood estimate is then a function of the number of samples drawn. The variance of this estimator is uniformly bounded, achieves the minimum variance for an unbiased estimator, and we can compute calibrated estimates of the variance. We provide theoretical arguments in favor of IBS and an empirical assessment of the method for maximum-likelihood estimation with simulation-based models. As case studies, we take three model-fitting problems of increasing complexity from computational and cognitive neuroscience. In all problems, IBS generally produces lower error in the estimated parameters and maximum log-likelihood values than alternative sampling methods with the same average number of samples. Our results demonstrate the potential of IBS as a practical, robust, and easy to implement method for log-likelihood evaluation when exact techniques are not available.

## Author summary

Researchers often validate scientific hypotheses by comparing data with the predictions of a mathematical or computational model. This comparison can be quantified by the

**Data Availability Statement:** Code and data to generate the figures in the paper is available at https://github.com/basvanopheusden/ibs-development. The IBS toolbox for efficient and

unbiased log-likelihood estimation is available at
https://github.com/lacerbi/ibs.

**Funding:** This work was supported by NSF grant
IIS-1344256 and NIH grant R01MH118925 to W.J.
M. The funders had no role in study design, data
collection and analysis, decision to publish, or
preparation of the manuscript.

**Competing interests:** The authors have declared
that no competing interests exist.

'log-likelihood', a number that captures how well the model explains the data. However,
for complex models common in neuroscience and computational biology, obtaining
exact formulas for the log-likelihood can be difficult. Instead, the log-likelihood is usu-
ally approximated by simulating synthetic observations from the model ('sampling'),
and seeing how often the simulated data match the actual observations. To reach correct
scientific conclusions, it is crucial that the log-likelihood estimates produced by such
sampling methods are accurate (unbiased). Here, we introduce inverse binomial sam-
pling (IBS), a method which differs from traditional approaches in that the number of
samples drawn from the model is not fixed, but adaptively adjusted in a simple way. For
each data point, IBS samples from the model until it matches the observation. We show
that IBS is unbiased and has other desirable statistical properties, both theoretically and
via empirical validation on three case studies from computational and cognitive neuro-
science. Across all examples, IBS outperforms fixed sampling methods, demonstrating
the utility of IBS as a practical, robust, and easy to implement method for log-likelihood
evaluation.

This is a *PLOS Computational Biology* Methods paper.

## Introduction

The *likelihood function* is one of the most important mathematical objects for modern statisti-
cal inference. Briefly, the likelihood function measures how well a model with a given set of
parameters can explain an observed data set. For a data set of discrete observations, the likeli-
hood has the intuitive interpretation of the probability that a random sample generated from
the model matches the data, for a given setting of the model parameters.

In many scientific disciplines, such as computational neuroscience and cognitive science,
computational models are used to give a precise quantitative form to scientific hypotheses and
theories. Statistical inference then plays at least two fundamental roles for scientific discovery.
First, our goal may be *parameter estimation* for a model of interest. Parameter values may have
a significance in themselves, for example we may be looking for differences in parameters
between distinct experimental conditions in a clinical or behavioral study. Second, we may be
considering a number of competing scientific hypotheses, instantiated by different models,
and we want to evaluate which model 'best' captures the data according to some criteria, such
as *explanation* (what evidence the data provide in favor of each model?) and *prediction* (which
model best predicts new observations?).

Crucially, the likelihood function is a key element for both parameter estimation and model
evaluation. A principled method to find best-fitting model parameters for a given data set is
maximum-likelihood estimation (MLE), which entails optimizing the likelihood function over
the parameter space [1]. Other common parameter estimation methods, such as maximum-a-
posteriori (MAP) estimation or full or approximate Bayesian inference of posterior distribu-
tions, still involve the likelihood function [2]. Moreover, almost all model comparison metrics
commonly used for scientific model evaluation are based on likelihood computations, from
predictive metrics such as Akaike's information criterion (AIC; [3]), the deviance information
criterion (DIC; [4]), the widely applicable information criterion (WAIC; [5]), leave-one-out
cross-validation [6]; to evidence-based metrics such as the marginal likelihood [7] and (loose)

approximations thereof, such as the Bayesian information criterion (BIC; [8]) or the Laplace approximation [7].

However, many complex computational models, such as those developed in computational biology [9–11], neuroscience [12, 13] and cognitive science [14], take the form of a generative model or *simulator*, that is an algorithm which given some context information and parameter settings returns one or more simulated observations (a synthetic data set). In those cases, the likelihood is often impossible to calculate analytically, and even when the likelihood might be available in theory, the numerical calculations needed to obtain it might be overwhelmingly expensive and intractable in practice. In such situations, the only thing one can do is to run the model to simulate observations ('samples'). In the absence of a likelihood function, common approaches to 'likelihood-free inference' generally try and match summary statistics of the data with summary statistics of simulated observations [15, 16].

In this paper, we ask instead the question of whether we can use samples from a simulator model to *directly* estimate the likelihood of the full data set, without recurring to summary statistics, in a 'correct' and 'efficient' manner, for some specific definition of these terms. The answer is *yes*, as long as we use the right *sampling method*.

In brief, a sampling method consists of a 'sampling policy' (a rule that determines how long to keep drawing samples for) and an 'estimator' which converts the samples to a real-valued number. To estimate the likelihood of a single observation (e.g., the response of a participant on a single trial of a behavioral experiment), the most obvious sampling policy is to draw a fixed amount of samples from the simulator model, and the simplest estimator is the fraction of samples that match the observation (or is 'close enough' to it, for continuous observations). However, most basic applications, such as computing the likelihood of multiple observations, require one to estimate the logarithm of the likelihood, or log-likelihood (see Methods for the underlying technical reasons). The 'fixed sampling' method described above cannot provide unbiased estimates for the log-likelihood (see Methods). Such bias vanishes in the asymptotic limit of infinite samples, but drawing samples from the model can be computationally expensive, especially if the simulator model is complex. In practice, the bias introduced by any fixed sampling method can translate to considerable biases in estimates of model parameters, or even reverse the outcome of model comparison analyses. In other words, using poor sampling methods can cause researchers to draw conclusions about scientific hypotheses which are not supported by their data.

In this work, we introduce *inverse binomial sampling* (IBS) as a powerful and simple technique for correctly and efficiently estimating log-likelihoods of simulator-based models. Crucially, IBS is a sampling method that provides uniformly unbiased estimates of the log-likelihood [17, 18] and calibrated estimates of their variance, which is also uniformly bounded.

We note that the problem of estimating functions $f(p)$ from observations of a Bernoulli distribution with parameter $p$ has been studied for mostly theoretical reasons in the mid-20th century, with major contributions represented by [17–21]. These works have largely focused on deriving the set of functions $f(p)$ for which an unbiased estimate exists, and demonstrating that for those functions, the inverse sampling policy (see Methods) is in a precise sense 'efficient'. Our main contribution here is to demonstrate that inverse binomial sampling provides a practically and theoretically efficient solution for a common problem in computational modeling; namely likelihood-free inference of complex models. To back up our claims, we provide theoretical arguments for the efficiency of IBS and a practical demonstration of its value for log-likelihood estimation and fitting of simulation-based models, in particular those used in computational cognitive science. We note that [22] had previously proposed inverse binomial sampling as a method for likelihood-free inference for certain econometric models,

but did not present an empirical assessment of the quality of the estimation and to our knowledge has not led to further adoption of IBS.

The paper is structured as follows. In the Methods, after setting the stage with useful "Definitions and notation", we describe more in detail the issues with the fixed sampling method and why they cannot be fixed ("Why fixed sampling fails"). We then present a series of arguments for why IBS solves these issues, and in particular why being unbiased here is of particular relevance ("Is inverse binomial sampling really better?"). In the Results, we present an empirical comparison of IBS and fixed sampling in the setting of maximum-likelihood estimation. As case studies, we take three model-fitting problems of increasing complexity from computational cognitive science: an 'orientation discrimination' task, a 'change localization' task, and a complex sequential decision making task. In all problems, IBS generally produces lower error in the estimated parameters than fixed sampling with the same average number of samples. IBS also returns solutions that are very close in value to the true maximum log-likelihood. We conclude by discussing further applications and extensions of IBS in the Discussion. Our theoretical analyses and empirical results demonstrate the potential of IBS as a practical, robust, and easy-to-implement method for log-likelihood evaluation when exact or numerical solutions are unavailable.

Implementations of IBS with tutorials and examples are available at the following link: https://github.com/lacerbi/ibs.

## Methods

### Definitions and notation

The two fundamental ingredients to run IBS are:

1. A data set $\mathcal{D} = \{(s_i, r_i)\}_{i=1}^{N}$ consisting of $N$ 'trials' characterized by 'stimuli' $s_i$ and *discrete* 'responses' $r_i$.

2. A generative model $g$ for the data (also known as a 'simulator'): a stochastic function that takes as input a stimulus $s$ and a parameter vector $\theta$ (and possibly other information) and outputs a response $r$.

In this section we expand on and provide motivations for the above assumptions, and introduce related definitions and notation used in the rest of the paper.

Here and in the following, for ease of reference, we use the language of behavioral and cognitive modeling (e.g., 'trial' for data points, 'stimulus' for independent or contextual variables, 'response' for observations or outcomes), but the statistical techniques that we discuss in the paper apply to any model and data set from any domain as long as they satisfy the fundamental assumption of IBS delineated above.

**The likelihood function.** We assume that we want to model a data set $\mathcal{D} = \{(s_i, r_i)\}_{i=1}^{N}$ consisting of $N$ 'trials' (data points), where

- $s_i$ is the *stimulus* (i.e., the experimental context, or independent variable) presented on the $i$-th trial; typically, $s_i$ is a scalar or vector of discrete or continuous variables (more generally, there are no restrictions on what $s_i$ can be as long as the simulator can accept it as input);

- $r_i$ is the *response* (i.e., the experimental observations, outcomes, or dependent variables) measured on the $i$-th trial; $r_i$ can be a scalar or vector, but crucially we assume it takes *discrete* values.

The requirement that $r_i$ be discrete will be discussed below.

Given a data set $\mathcal{D}$, and a model parametrized by parameter vector $\boldsymbol{\theta}$, we can write the *likelihood function* for the responses given the stimuli and model parameters as

$$
\begin{aligned}
\Pr(\{\boldsymbol{r}_i\}_{i=1}^N | \{\boldsymbol{s}_i\}_{i=1}^N, \boldsymbol{\theta}) &= \prod_{i=1}^N \Pr(\boldsymbol{r}_i | \boldsymbol{r}_1, \ldots, \boldsymbol{r}_{i-1}, \boldsymbol{s}_1, \ldots, \boldsymbol{s}_N, \boldsymbol{\theta}) \\
&= \prod_{i=1}^N \Pr(\boldsymbol{r}_i | \boldsymbol{r}_1, \ldots, \boldsymbol{r}_{i-1}, \boldsymbol{s}_1, \ldots, \boldsymbol{s}_i, \boldsymbol{\theta}),
\end{aligned}
\tag{1}
$$

where the first line follows from the chain rule of probability, and holds in general, whereas in the second step we applied the reasonable 'causal' (or 'no-time-travel') assumption that the response at the $i$-th trial is not influenced by future stimuli. We also used the causality assumption that current responses are not influenced by future responses to choose a specific order to apply the chain rule in the first line.

Note that Eq 1 assumes that the researcher is not interested in statistically modeling the stimuli, which are taken to be given (i.e., on the right-hand side of the conditional probability). This choice is without loss of generality, as any variable of statistical interest can always be relabeled and become an element of the 'response' vector. For compactness, from now on we will denote $\Pr(\{\boldsymbol{r}_i\}_{i=1}^N | \{\boldsymbol{s}_i\}_{i=1}^N, \boldsymbol{\theta}) \equiv \Pr(\mathcal{D}|\boldsymbol{\theta})$, in a slight abuse of notation.

Eq 1 describes the most general class of models, in which the response in the current trial might be influenced by the history of both previous stimuli and previous responses. Many models commonly make a stronger conditional independence assumption between trials, such that the response on the current trial only depends on the current stimulus. Under this stronger assumption, the likelihood takes a simpler form,

$$
\Pr(\mathcal{D}|\boldsymbol{\theta}) = \prod_{i=1}^N \Pr(\boldsymbol{r}_i | \boldsymbol{s}_i, \boldsymbol{\theta}).
\tag{2}
$$

While Eq 2 is simpler, it still includes a wide variety of models. For example, note that time-dependence can be easily included in the model by incorporating time into the 'stimulus' $\boldsymbol{s}$, and including time-dependent parameters explicitly in the model specification. In the rest of this work, for simplicity we consider models that make conditional independence assumptions as in Eq 2, but our techniques apply in general also for likelihoods as per Eq 1.

Given that the likelihood of the $i$-th trial can be directly interpreted as the probability of observing response $\boldsymbol{r}_i$ in the $i$-th trial (conditioned on everything else), we denote such quantity with $p_i \in [0, 1]$. The value $p_i$ is a function of $\boldsymbol{\theta}$, depends on the current stimulus and response, and may or may not depend on previous stimuli or responses.

With this notation, we can simply write the likelihood as

$$
\Pr(\mathcal{D}|\boldsymbol{\theta}) = \prod_{i=1}^N p_i.
\tag{3}
$$

Finally, we note that it is common practice to work with the logarithm of the likelihood, or log-likelihood, that is

$$
\mathcal{L}(\boldsymbol{\theta}) \equiv \log \Pr(\mathcal{D}|\boldsymbol{\theta}) = \log \prod_{i=1}^N p_i = \sum_{i=1}^N \log p_i.
\tag{4}
$$

The typical rationale for switching to the log-likelihood is that for large $N$ the likelihood tends to be a vanishingly small quantity, so the logarithm makes it easier to handle numerically

('numerical convenience'). However, we will see later that there are statistically meaningful reasons to prefer the logarithmic representation (i.e., a sum of independent terms).

Crucially, we assume that the likelihood function is unavailable in a tractable form—for example, because the model is too complex to derive an analytical expression for the likelihood. Instead, IBS provides a technique for estimating Eq 4 via simulation.

**The generative model or simulator.** While we assume no availability of an *explicit* representation of the likelihood function, we assume that the model of interest is represented *implicitly* by a stochastic generative model (or 'simulator'). In the most general case, the simulator is a stochastic function $g$ that takes as input the current stimulus $s_i$, arrays of past stimuli and responses, and a parameter vector $\theta$, and outputs a discrete response $r_i$, *conditional* on all past events,

$$r_i \sim g(s_1, \ldots, s_i, r_1, \ldots, r_{i-1}; \theta). \tag{5}$$

As mentioned in the previous section, a common assumption for a model is that the response in the current trial only depends on the current stimulus and parameter vector, in which case

$$r \sim g(s; \theta). \tag{6}$$

For example, the model $g(\cdot)$ could be simulating the responses of a human participant in a complex cognitive task; the (discrete) choices taken by a rodent in a perceptual decision-making experiment; or the spike count of a neuron in sensory cortex for a specific time bin after a stimulus presentation.

We list now the requirements that the simulator model needs to satisfy to be used in conjuction with IBS.

- *Discrete response space.* Lacking an expression for the likelihood function, the only way to estimate the likelihood or any function thereof is by drawing samples $r \sim g(s_i, \ldots; \theta)$ on each trial, and matching them to the response $r_i$. This approach requires that there is a nonzero probability for a random sample $r$ to match $r_i$, hence the assumption that the space of responses is discrete. We will see in the Discussion a possible method to extend IBS to larger or continuous response spaces.

- *Conditional simulation.* An important requirement of the generative model, stated implicitly by Eqs 5 and 6, is that the simulator should afford *conditional simulation*, in that we can simulate the response $r_i$ for any trial $i$, given the current stimulus $s_i$, and possibly previous stimuli and responses. Note that this class of models, while large, does not include *all* possible simulators, in that some simulators might not afford conditional generation of responses. For example, models with latent dynamics might be able to generate a full sequence of responses given the stimuli, but it might not be easy or computationally tractable to generate the response in a given trial, conditional on a specific sequence of previous responses.

- *Computational cost.* Finally, for the purpose of some of our analyses we assume that drawing a sample from the generative model is at least moderately computationally expensive, which limits the approximate budget of samples one is willing to use for each likelihood evaluation (in our analyses, up to about a hundred, on average, per likelihood evaluation). Number of samples is a reasonable proxy for any realistic resource expenditure since most costs (e.g., time, energy, number of processors) would be approximately proportional to it. Therefore, we also require that every response value in the data has a non-negligible probability of being sampled from the model—given the available budget of samples one can reasonably draw. In this paper, we will focus on the low-sample regime, since that is where IBS considerably outperforms other approaches.

For our analyses of performance of the algorithm, we also assume that the computational cost is independent of the stimulus, response or model parameters, but this is not a requirement of the method.

**Reduction to Bernoulli sampling.** Given the conditional independence structure codified by Eq 3, to estimate the log-likelihood of the entire data set, we cannot do better than estimating $p_i$ on each trial independently, and combining the results. However, combining estimates $\hat{p}_i$ into a well-behaved estimate of $\prod_{i=1}^{N} p_i$ is non-trivial (see "Why not an unbiased estimator of the likelihood?"). Instead, it is easier to estimate $\mathcal{L}_i \equiv \log p_i$ for each trial and calculate the log-likelihood

$$\mathcal{L}(\boldsymbol{\theta}) = \log \Pr(\mathcal{D}|\boldsymbol{\theta}) = \sum_{i=1}^{N} \log p_i = \sum_{i=1}^{N} \mathcal{L}_i. \tag{7}$$

We can estimate this log-likelihood by simply summing estimates $\hat{\mathcal{L}}_i$ across trials, in which case the central limit theorem guarantees that the distribution of $\hat{\mathcal{L}}(\boldsymbol{\theta})$ is normally distributed for large values of $N$, which is true for typical values of $N$ of the order of a hundred or more (see later).

We can make one additional simplification, without loss of generality. The generative model specifies an implicit probability distribution $r_i \sim g(s_i, \ldots; \boldsymbol{\theta})$ for each trial. However, to estimate the log-likelihood, we do not need to know the full distribution, only the probability for a random sample $\boldsymbol{r}$ from the model to match the observed response $\boldsymbol{r}_i$. Therefore, we can convert each sample $\boldsymbol{r}$ to

$$x = \begin{cases} 1 & \text{if } \boldsymbol{r} = \boldsymbol{r}_i & \text{('hit')} \\ 0 & \text{otherwise} & \text{('miss')}, \end{cases} \tag{8}$$

and lose no information relevant for estimating the log-likelihood. By construction, $x$ follows a Bernoulli distribution with probability $p_i$. Note that this holds regardless of the type of data, the structure of the generative model or the model parameters. The only difference between different models and data sets is the distribution of the likelihood $p_i$ across trials. Moreover, since $p_i$ is interpreted as the parameter of a Bernoulli distribution, we can apply standard frequentist or Bayesian statistical reasoning to it.

In conclusion, we can reduce the problem of estimating the log-likelihood of a given model by sampling to a smaller problem: given a method to draw samples $(x_1, x_2, \ldots)$ from a Bernoulli distribution with unknown parameter $p$, estimate $\log p$ as precisely and accurately as possible using on average as few samples as possible.

**Sampling policies and estimators.** A *sampling policy* is a function that, given a sequence of samples $\boldsymbol{x} \equiv (x_1, x_2, \ldots, x_k)$, decides whether to draw an additional sample or not [20]. In this work, we compare two sampling policies:

1. The commonly used *fixed* policy: Draw a fixed number of samples $M$, then stop.

2. The *inverse binomial sampling* policy: Keep drawing samples until $x_k = 1$, then stop.

In our case, an *estimator* (of $\log p$) is a function $\hat{\mathcal{L}}(\boldsymbol{x})$ that takes as input a sequence of samples $\boldsymbol{x} = (x_1, x_2, \ldots, x_k)$ and returns an estimate of $\log p$. We recall that the *bias* of an estimator $\hat{\mathcal{L}}$ of $\log p$, for a given true value of the Bernoulli parameter $p$, is defined as

$$\text{Bias}[\hat{\mathcal{L}}|p] = \mathbb{E}[\hat{\mathcal{L}}] - \log p, \tag{9}$$

where the expectation is taken over all possible sequences $\boldsymbol{x}$ generated by the chosen sampling

policy under the Bernoulli probability $p$. Such estimator is (uniformly) *unbiased* if $\text{Bias}[\hat{\mathcal{L}}|p] = 0$ for all $0 < p \leq 1$ (that is, the estimator is centered around the true value).

- *Fixed sampling.* For the fixed sampling policy, since all samples are independent and identically distributed, a sufficient statistic for estimating $p$ from the samples $(x_1, x_2, \ldots, x_M)$ is the number of 'hits', $m(\boldsymbol{x}) \equiv \sum_{k=1}^{M} x_k$. The most obvious estimator for an obtained sequence of samples $\boldsymbol{x}$ is then

$$\hat{\mathcal{L}}_{\text{naive}}(\boldsymbol{x}) = \log\left(\frac{m(\boldsymbol{x})}{M}\right), \tag{10}$$

but this estimator has infinite bias; since as long as $p \neq 1$, there is always a nonzero chance that $m(\boldsymbol{x}) = 0$, in which case $\hat{\mathcal{L}}_{\text{naive}}(\boldsymbol{x}) = -\infty$ (and thus $\mathbb{E}[\hat{\mathcal{L}}_{\text{naive}}] = -\infty$). This divergence can be fixed in multiple ways; in the main text we use

$$\hat{\mathcal{L}}_{\text{fixed}}(\boldsymbol{x}) = \log\left(\frac{m(\boldsymbol{x}) + 1}{M + 1}\right). \tag{11}$$

Note that *any* estimator based on the fixed sampling policy will always produce biased estimates of $\log p$, as guaranteed by the reasoning in the following section. As an empirical validation, we show in S1 Appendix B.1 that our results do not depend on the specific choice of estimator for fixed sampling (Eq 11).

- *Inverse binomial sampling.* For inverse binomial sampling we note that, since $x$ is a binary variable, the policy will always result in a sequence of samples of the form

$$\boldsymbol{x} = (\overbrace{0, 0, 0, 0, 0, \ldots, 0, 1}^{K}), \tag{12}$$

where the length of the sequence is a stochastic variable, which we label $K$ (a positive integer). Moreover, since each sample is independent and a 'hit' with probability $p$, the length $K$ follows a geometric distribution with parameter $1 - p$,

$$\Pr(K = k) = p(1 - p)^{k-1}. \tag{13}$$

We convert a value of $K$ into an estimate for $\log p$ using the IBS estimator,

$$\hat{\mathcal{L}}_{\text{IBS}}(\boldsymbol{x}) = \begin{cases} 0 & \text{for } K = 1 \\ -\sum_{k=1}^{K-1} \frac{1}{k} & \text{for } K > 1. \end{cases} \tag{14}$$

Crucially, Eq 14 combined with the IBS policy provides a uniformly unbiased estimator of $\log p$ [18]. Moreover, we can show that $\hat{\mathcal{L}}_{\text{IBS}}$ is the *uniformly minimum-variance unbiased estimator* of $\log p$ under the IBS policy. For a full derivation of the properties of the IBS estimator, we refer to S1 Appendix A.1. Eq 14 can be written compactly as $\hat{\mathcal{L}}_{\text{IBS}}(K) = \psi(1) - \psi(K)$, where $\psi(z)$ is the *digamma function* [23].

We now provide an understanding of why fixed sampling is not a good policy, despite its intuitive appeal, and then show why IBS solves many of the problems with fixed sampling.

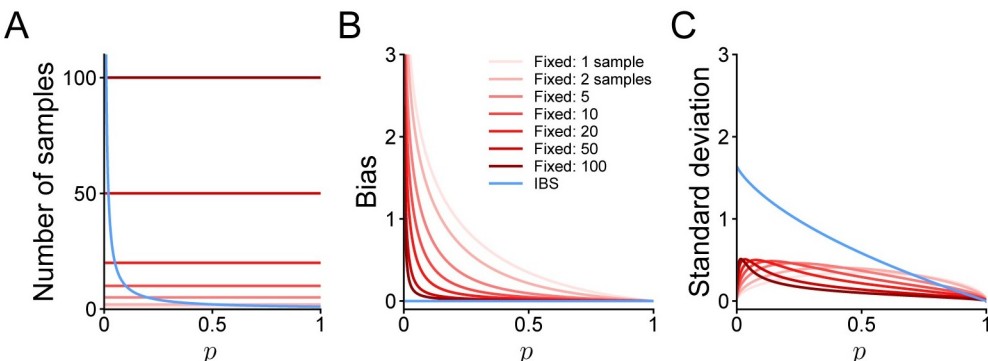

**Fig 1. Properties of estimators. A**. Number of samples used by fixed (red curves) or inverse binomial sampling (blue; expected value) to estimate the log-likelihood log $p$ on a single trial with probability $p$. IBS uses on average $\frac{1}{p}$ trials. **B**. Bias of the log-likelihood estimate. The bias of IBS is identically zero. **C**. Standard deviation of the log-likelihood estimate.

## Why fixed sampling fails

We summarize in Fig 1 the properties of the IBS estimator and of fixed sampling, for different number of samples $M$, as a function of the trial likelihood $p$. In particular, we plot the expected number of samples, the bias, and the standard deviation of the estimators.

The critical disadvantage of the fixed sampling policy with $M$ samples is that its estimates of the log-likelihood are inevitably biased (see Fig 1B). Fixed sampling is 'inevitably' biased because the bias decreases as one takes more samples, but for $p \to 0$, the estimator remains biased. More precisely, in a joint limit where $M \to \infty$, $p \to 0$ and $pM \to \lambda$ for some constant $\lambda$, the bias collapses onto a single 'master curve' (see Fig 2; and S1 Appendix A.2 for the derivation). In particular, we observe that the bias is close to zero for $\lambda \gg 1$ and that it diverges when $\lambda \ll 1$, or equivalently, for $M \gg \frac{1}{p}$ and $M \ll \frac{1}{p}$, respectively.

To convey the intuition for why the bias diverges for small probabilities, we provide a gambling analogy. Imagine playing a slot machine and losing the first 100 bets you make. You can now deduce that this slot machine likely has a win rate less than 1%, but there is no way of knowing whether it is 1%, 0.1%, 0.01% or even 0% apart from any prior beliefs you may have (for example, you expect that the house has stacked the odds in their favor but not

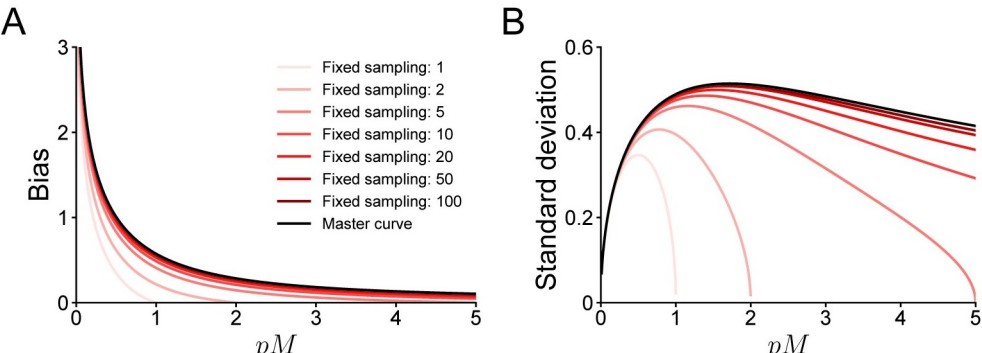

**Fig 2. Asymptotic properties of fixed sampling. A**. Bias of fixed sampling estimators of the log-likelihood, plotted as a function of $pM$, where $p$ is the likelihood on a given trial, and $M$ the number of samples. As $M \to \infty$, the bias converges to a master curve (Eq S3 in S1 Appendix). **B**. Same, but for standard deviation of the estimate.

overwhelmingly so). In practice, this uncertainty is unlikely to affect your decision whether to continue playing the slot machine, since the expected value of the slot machine depends linearly on its win rate. However, if your goal is to estimate the *logarithm* of the win rate, the difference between these percentages becomes infinitely large as the true win rate tends to 0. We provide a more formal treatment of the bias of fixed sampling in S1 Appendix A2.

**Why fixed sampling cannot be fixed.** The asymptotic analyses above suggest an obvious solution to prevent fixed sampling estimators from becoming strongly biased: make sure to draw enough samples so that $M \gg \max_{i=1...N} \frac{1}{p_i}$. Although this solution will succeed in theory, it has practical issues. First of all, choosing $M$ requires knowledge of $p_i$ on each trial, which is equivalent to the problem we set out the solve in the first place. Moreover, even if one can derive or estimate an upper bound on $\frac{1}{p_i}$ (for example, in behavioral models that include a lapse rate, that is a nonzero probability of giving a uniformly random response), fixed sampling will be inefficient. As shown in Fig 2, the bias in $\hat{\mathcal{L}}_{\text{fixed}}$ is small when $\lambda \approx 1$ or $M \approx \frac{1}{p}$ and increasing $M$ even further has diminishing returns, at least for the purpose of reducing bias. If we choose $M$ inversely proportional to the probability $p_i$ on the trial where the model is least likely to match the observed response, we will draw many more samples than necessary for all other trials.

One might hope that in practice the likelihood $p_i$ is approximately the same across trials, but the opposite is true. As an example, take a typical 'orientation discrimination' psychophysical task in which a participant has to detect whether a presented oriented grating is tilted clockwise or anti-clockwise from vertical, and consider a generative model for the observer's responses that includes sensory measurement noise and lapses (see Results for details). Moreover, imagine that the experiment contains $\approx$ 500 trials, and the participant's true lapse rate is 1%. The model will always assign more probability to correct responses than errors, so, for all correct trials, $p_i$ will be at least 0.5. However, there will likely be a handful of trials where the participant lapses and makes a grave error (responding incorrectly to a stimulus very far from the decision boundary), in which case $p_i$ will be 0.5 times the lapse rate. This hypothetical scenario is not exceptional, in fact it is almost inevitable in any experiment where participants occasionally make unpredictable responses, and perform hundreds or more trials.

A more sophisticated solution would relax the assumption that $M$ needs to be constant for all trials, and instead choose $M$ as a function of $p_i$ on each trial. However, since $p_i$ is unknown, one would need to first estimate $p_i$ by sampling, choose $M_i$ for each trial, then re-estimate $\mathcal{L}_i$. Such an iterative procedure would create a non-fixed sampling scheme, in which $M_i$ adapts to $p_i$ on each trial. This approach is promising, and it is, in fact, how we originally arrived at the idea of using inverse binomial sampling for log-likelihood estimation, while working on the complex cognitive model described in the Results.

Finally, a heuristic solution would be to disregard any statistical concerns, pick $M$ based on some intuition or from similar studies in the literature, and hope that the bias turns out to be negligible. We do not intend to dissuade researchers from using such pragmatic approaches if they work in practice. Unfortunately, this one does not. As Fig 2 shows, estimating log-likelihoods with fixed sampling can cause biases of 1 or more points of model evidence if the data set contains even a single trial on which $p_i \leq \frac{1}{2M}$. Since differences in log-likelihoods larger than 5 to 10 points are often regarded as strong evidence for one model over another [24–26], it is well possible for such biases to reverse the outcome of a model comparison analysis. This point bears repeating; if one uses fixed sampling to estimate log-likelihoods and the number of samples is too low, one risks of drawing conclusions about scientific hypotheses that are not supported by the experimental data one has collected.

**Why not an unbiased estimator of the likelihood?** In this paper, we focus on finding an unbiased estimator of the log-likelihood, but one might wonder why we do not look instead for an unbiased estimator of the likelihood. In fact, we already have such an estimator. Fixed sampling provides an unbiased estimate of $p_i$ for each trial, and since all estimates are unbiased and statistically independent, $\prod_i p_i$ is also unbiased.

The critical issue is the shape of the distribution of these estimates. While a central limit theorem for products of random variables exists, it only holds if all the estimates are almost surely not zero. For estimates of $p_i$ obtained via fixed sampling, this is not the case, and we would not obtain a well-behaved (i.e., log-normal) distribution of $\prod_i p_i$. In fact, the distribution would be highly multimodal with the main peak being at $\prod_i p_i = 0$. This property makes this estimator unusable for all practical purposes (e.g., from maximum-likelihood estimation to Bayesian inference).

Instead, by switching to log-likelihood estimation, we can find an estimator (IBS) which is both unbiased and whose estimates are guaranteed to be well-behaved (in particular, normally distributed). We stress that normality is not just a desirable addition, but a fundamental feature with substantial practical consequences for how estimators are used, as we will see more in detail in the following section.

## Is inverse binomial sampling really better?

While one could expect that the unbiasedness of the IBS estimator would come at a cost, such as more samples, a much higher variance, or perhaps a particularly complex implementation, we show here that IBS is not only unbiased, but it is sample-efficient, its estimates are low-variance, and can be implemented in a few lines of code.

**Implementation.** We present in Table 1 a description in pseudo-code of the basic IBS algorithm to estimate the log-likelihood of a given parameter vector $\boldsymbol{\theta}$ for a given data set and generative model. The procedure is based on the inverse binomial sampling scheme introduced previously, generalized sequentially to multiple trials.

For each trial, we draw sampled responses from the generative model, given the stimulus $\boldsymbol{s}_i$ in that trial, using the subroutine `sample_from_model`, until one matches the observed response $\boldsymbol{r}_i$. This yields a value of $K_i$ on each trial $i$, which IBS converts to an estimate $\hat{\mathcal{L}}_i$ (where we use the convention that a sum with zero terms equals 0). We make our way sequentially across all trials, returning then the summed log-likelihood estimate $\hat{\mathcal{L}}_{\text{IBS}}$ for the entire data set.

**Table 1. Inverse binomial sampling algorithm.**

**Input**: Stimuli $\{\boldsymbol{s}_i\}_{i=1}^N$, responses $\{\boldsymbol{r}_i\}_{i=1}^N$, generative model $\mathcal{M}$, parameters $\boldsymbol{\theta}$

| | |
|---|---|
| 1: **for** $i \leftarrow 1 \dots N$ **do** | ▷ Sequential loop over all trials |
| 2: $K_i \leftarrow 1$ | |
| 3: **while** `sample_from_model`$(\mathcal{M}, \boldsymbol{\theta}, \boldsymbol{s}_i) \neq \boldsymbol{r}_i$ **do** | |
| 4: $K_i \leftarrow K_i + 1$ | |
| 5: $\hat{\mathcal{L}}_i \leftarrow -\sum_{k=1}^{K_i-1} \frac{1}{k}$ | ▷ IBS estimator from Eq 14 |
| 6: **return** $\sum_{i=1}^{N} \hat{\mathcal{L}}_i$ | ▷ Return total log-likelihood estimate |

Basic sequential implementation of the IBS estimator.

In practice, depending on the programming language of choice, it might be useful to take advantage of numerical features such as vectorization to speed up computations. An alternative 'parallel' implementation of IBS is described in S1 Appendix.

One might wonder how to choose the $\theta$ to evaluate in the algorithm in Table 1 in the first place. IBS is agnostic of how candidate $\theta$ are proposed. Most often, the function that implements the IBS algorithm will be passed to a chosen optimization or inference algorithm, and the job of proposing $\theta$ will be taken care of by the chosen method. For maximum-likelihood estimation, we recommend derivative-free optimization methods that deal effectively and robustly with noisy evaluations, such as Bayesian Adaptive Direct Search (BADS; [27]) or noise-robust CMA-ES [28, 29]. At the end of optimization, most methods will then return a candidate solution $\hat{\theta}_{\mathrm{MLE}}$ and possibly an estimate of the value of the target function at $\hat{\theta}_{\mathrm{MLE}}$. However, since the target function is noisy and the final estimate is biased (because, by definition, it is better than the other evaluated locations), we recommend to re-estimate $\hat{\mathcal{L}}_{\mathrm{IBS}}(\hat{\theta}_{\mathrm{MLE}})$ multiple times to obtain a higher-precision, unbiased estimate of the log-likelihood at $\hat{\theta}_{\mathrm{MLE}}$ (see also later, "Iterative multi-fidelity").

Implementations of IBS in different programming languages can be found at the following web page: https://github.com/lacerbi/ibs.

**Computational time.** The number of samples that IBS takes on a trial with probability $p_i$ is geometrically distributed with mean $\frac{1}{p_i}$. We saw earlier that for fixed-sampling estimators to be approximately unbiased, one needs at least $\frac{1}{p_i}$ samples, and IBS does exactly that in expectation. Moreover, since IBS adapts the number of samples it takes on different trials, it will be considerably more sample-efficient than fixed sampling with constant $M$ across trials. For example, in the aforementioned example of the orientation discrimination task, when most trials have a likelihood $p_i \geq 0.5$, IBS will often take just 1 or 2 samples on those trials. Therefore, it will allocate most of its samples and computational time on trials where $p_i$ is low and those samples are needed.

**Variance.** The variance of the IBS estimator can be derived as

$$\mathrm{Var}\left[\hat{\mathcal{L}}_{\mathrm{IBS}}\right] = \sum_{k=1}^{\infty} \frac{1}{k^2}(1-p)^k = \mathrm{Li}_2(1-p), \tag{15}$$

where we introduced the dilogarithm or Spence's function $\mathrm{Li}_2(z)$ [30]. The variance (plotted in Fig 1C as standard deviation) increases when $p \to 0$, but it does not diverge; instead, it converges to $\frac{\pi^2}{6}$. Therefore, IBS is not only uniformly unbiased, but its variance is uniformly bounded. The full derivation of Eq 15 is reported in S1 Appendix A.3.

We already mentioned that $\hat{\mathcal{L}}_{\mathrm{IBS}}$ is the minimum-variance unbiased estimator of $\log p$ given the inverse binomial sampling policy, but it also comes *close* (less than $\sim 30\%$ distance) to saturating the *information inequality*, which specifies the minimum variance that can be theoretically achieved by any estimator under a non-fixed sampling policy (an analogue of the Cramer-Ráo bound [18]). We note that fixed sampling eventually saturates the information inequality in the limit $M \to \infty$, but as mentioned in the previous section, the fixed-sampling approach can be highly wasteful or substantially biased (or both), not knowing a priori how large $M$ has to be across trials. See S1 Appendix A.4 for a full discussion of the information inequality and comparison between estimators.

Eq 15 has theoretical relevance, but requires us to know the true value of the likelihood $p$, which is unknown in practice. Instead, we define the estimator of the variance of a specific IBS

estimate, having sampled for $K$ times until a 'hit', as

$$\mathrm{Var}[\hat{\mathcal{L}}_{\mathrm{IBS}}|K] = \psi_1(1) - \psi_1(K), \tag{16}$$

where $\psi_1(z)$ is the *trigamma function*, the derivative of the digamma function [23]. We derived Eq 16 from a Bayesian interpretation of the IBS estimator, which can be found in S1 Appendix A.5. Note that Eqs 15 and 16 correspond to slightly different concepts, in that the former represents the variance of the estimator for a known $p$ (from a frequentist point of view), while the latter is the posterior variance of $\mathcal{L}$ for a known $K$ (for which there is no frequentist analogue). See also later "Higher-order moments" for further discussion.

**Iterative multi-fidelity.**   We define a *multi-fidelity* estimator as an estimator with a tunable parameter that affords different degrees of precision at different computational costs (i.e., from a cheaper, inaccurate estimate to a very accurate but expensive one), borrowing the term from the literature on computer simulations and surrogate models [31, 32]. IBS provides a particularly convenient way to construct an *iterative* multi-fidelity estimator in that we can perform $R$ independent 'repeats' of the IBS estimate at $\boldsymbol{\theta}$, and combine them by averaging,

$$
\begin{aligned}
\hat{\mathcal{L}}_{\mathrm{IBS}-R}(\boldsymbol{\theta}) &= \frac{1}{R}\sum_{r=1}^{R}\hat{\mathcal{L}}_{\mathrm{IBS}}^{(r)}(\boldsymbol{\theta}) \\
\mathrm{Var}[\hat{\mathcal{L}}_{\mathrm{IBS}-R}(\boldsymbol{\theta})] &= \frac{1}{R^2}\sum_{r=1}^{R}\mathrm{Var}\left[\hat{\mathcal{L}}_{\mathrm{IBS}}^{(r)}(\boldsymbol{\theta})\right],
\end{aligned}
\tag{17}
$$

where $\hat{\mathcal{L}}_{\mathrm{IBS}}^{(r)}$ denotes the $r$-th independent estimate obtained via IBS. For $R = 1$, we recover the standard ('1-rep') IBS estimator. The variances in Eq 17 are computed empirically using the estimator in Eq 16.

Importantly, we do not need to perform all $R$ repeats at the same time, but we can iteratively refine our estimates whenever needed, and only need to store the current estimate, its variance and the number of repeats performed so far:

$$
\begin{aligned}
\hat{\mathcal{L}}_{\mathrm{IBS}-R+1}(\boldsymbol{\theta}) &= \frac{1}{R+1}\left[R\cdot\hat{\mathcal{L}}_{\mathrm{IBS}-R}(\boldsymbol{\theta}) + \hat{\mathcal{L}}_{\mathrm{IBS}}^{(r+1)}(\boldsymbol{\theta})\right] \\
\mathrm{Var}[\hat{\mathcal{L}}_{\mathrm{IBS}-R+1}(\boldsymbol{\theta})] &= \frac{1}{(R+1)^2}\left\{R^2\cdot\mathrm{Var}\left[\hat{\mathcal{L}}_{\mathrm{IBS}-R}(\boldsymbol{\theta})\right] + \mathrm{Var}\left[\hat{\mathcal{L}}_{\mathrm{IBS}}^{(r+1)}(\boldsymbol{\theta})\right]\right\}.
\end{aligned}
\tag{18}
$$

Crucially, while a similar procedure could be performed with any estimator (including fixed sampling), the fact that IBS is unbiased and its variance is bounded ensures that the combined iterative estimator is also unbiased and eventually converges to the true value for $R \to \infty$, with variance bounded above by $\frac{\pi^2}{6R}$.

Finally, we note that the iterative multi-fidelity approach described in this section can be extended such that, instead of having the same number of repeats $R$ for all trials, one could adaptively allocate a different number of repeats $r_i$ to each trial so as to minimize the overall variance of the estimated log-likelihood (see S1 Appendix C.2).

**Bias or variance?**   In the previous sections, we have seen that IBS is always unbiased, whereas fixed sampling can be highly biased when using too few samples. However, with the right choice of $M$, fixed sampling can have lower variance. We now list several practical and theoretical arguments for why bias can have a larger negative impact than variance, and being unbiased is a desirable property for an estimator of the log-likelihood.

1. To use IBS or fixed sampling to estimate the log-likelihood of a given data set, we sum estimates of $\mathcal{L}_i$ across trials. Being the sum of independent random variables, as $N \to \infty$, the standard deviation of $\hat{\mathcal{L}}(\boldsymbol{\theta})$ will grow proportional to $\sqrt{N}$, whereas the bias grows linearly with $N$. For a concrete example, see S1 Appendix A.6.

2. When using the log-likelihood (or a derived metric) for model selection, it is common to collect evidence for a model, possibly hierarchically, across multiple datasets (e.g., different participants in a behavioral experiment), which provides a second level of averaging that can reduce variance but not bias.

3. Besides model selection, the other key reason to estimate log-likelihoods is to infer parameters of a model, for example via maximum-likelihood estimation. For this purpose, one would use an optimization algorithm that calls the routine that estimates $\hat{\mathcal{L}}(\boldsymbol{\theta})$ many times with different candidate values of $\boldsymbol{\theta}$, and uses this information to estimate the value that maximizes $\mathcal{L}(\boldsymbol{\theta})$. Powerful, sample-efficient optimization algorithms, such as those based on Bayesian optimization, work by building a statistical approximation (a *surrogate*) of the objective function [27, 33–35], most commonly via Gaussian processes [36]. These methods can operate successfully with noisy objectives by effectively averaging function values from nearby parameter vectors. By contrast, no optimization algorithm can handle bias. This argument is not limited to maximum-likelihood estimation, as recent methods have been proposed to use Gaussian process surrogates to perform (approximate) Bayesian inference and infer posterior distributions [37–40]; also these techniques can handle variance in the estimates but not bias.

4. The ability to combine unbiased estimates of known variance iteratively (as described in the previous section) is particularly useful with adaptive fitting methods based on Gaussian processes, whose algorithmic cost grows super-linearly in the number of *distinct* training points [36]. Thanks to iterative multi-fidelity estimation, these methods would have the opportunity to refine their estimates of the log-likelihood at a previously evaluated point, whenever deemed useful, without incurring an increased algorithmic cost.

5. On a conceptual level, bias is potentially more dangerous than variance. Bias can cause researchers to confidently draw false conclusions, variance, when properly accounted for, causes decreased statistical power and lack of confidence. Appropriate statistical tools can account for variance and explain seemingly conflicting findings resulting from underpowered studies [41], whereas bias is much harder to recognize or correct no matter what statistical techniques one uses.

Finally, while for the sake of argument we structured this section as an opposition between bias and variance, it is not an exact dichotomy and both properties matter, together with other considerations (see S1 Appendix A.6). For example, there are situations in which the researcher may knowingly decide to increase bias to reduce both variance and computational costs. Notably, while this trade-off is easy to achieve with the IBS estimator (see S1 Appendix C.1), there is no similarly easy technique to de-bias a biased estimator.

**Higher-order moments.**   So far, we have considered the mean (or bias) and variance of $\hat{\mathcal{L}}_{\text{fixed}}$ and $\hat{\mathcal{L}}_{\text{IBS}}$ in detail, but ignored any higher-order moments. This is justified since to estimate the log-likelihood of a model with a given parameter vector we will sum these estimates across many trials. Therefore, the central limit theorem guarantees that the distribution of $\mathcal{L}(\boldsymbol{\theta})$ is Gaussian with mean and variance determined by the mean and variance of $\hat{\mathcal{L}}_{\text{fixed}}$ or $\hat{\mathcal{L}}_{\text{IBS}}$ on each trial, at least as long as the distribution of $p_i$ across trials satisfies a regularity

condition (specifically, the Lindeberg [42] or Lyapunov conditions [43, Chapter 7.3], both of which place restrictions on the degree to which the variance of any single trial can dominate the distribution of $\sum_i \hat{\mathcal{L}}_i$). A sufficient but far from necessary condition is that there exists a lower bound on $p_i$, which is the case for example for a behavioral model with a lapse rate. Using the same argument, the total number of samples $K_{\text{tot}}$ that IBS uses to estimate $\mathcal{L}(\theta)$ is also approximately Gaussian.

In the following, we demonstrate empirically that the distributions of the number of samples taken by IBS and of the estimates $\hat{\mathcal{L}}_{\text{IBS}}$ are Gaussian. Importantly, we also show that the estimate of the variance from Eq 16, $\hat{V}_{\text{IBS}}$, is *calibrated*. That is, we expect the fraction of estimates within the credible interval $\hat{\mathcal{L}}_{\text{IBS}} \pm \beta \sqrt{\hat{V}_{\text{IBS}}}$ to be (approximately) $\Phi(\beta) - \Phi(-\beta)$, where $\Phi(x)$ is the cumulative normal distribution function and $\beta > 0$.

As a realistic scenario, we consider the psychometric function model described in the Results. For each simulated data set, we estimated the log-likelihood under the true data-generating parameters $\theta_{\text{true}}$ (see S1 Appendix B.1 for details). Specifically, for each data set we ran IBS and recorded the estimated log-likelihood $\hat{\mathcal{L}}_{\text{IBS}}$, the total number of samples $K_{\text{tot}}$ taken, and a Bayesian estimate for the variance of $\hat{\mathcal{L}}_{\text{IBS}}$ from Eq 16. For the total number of samples $K_{\text{tot}}$ and the $\hat{\mathcal{L}}_{\text{IBS}}$ estimate, we can compute the theoretical mean and variance by knowing the trial likelihoods $p_i$, which we can evaluate exactly in this example.

For each obtained $K_{\text{tot}}$, we computed a $z$-score by subtracting the exact mean and dividing by the exact standard deviation, obtained by knowing the mean and variance of geometric random variables underlying the samples taken in each trial. If $K_{\text{tot}}$ is normally distributed, we expect that the variable $z$ across data sets should appear to be distributed as a standard normal, $z \sim \mathcal{N}(0, 1)$. If $K_{\text{tot}}$ is not normally distributed, we should see deviations from normality in the distribution of $z$, especially in the tails. By comparing the histogram of $z$-scores with a standard normal in Fig 3A, we see that the total number of samples is approximately normal, with some residual skew.

We did the same analysis for the estimate $\hat{\mathcal{L}}_{\text{IBS}}$, using the $z$-scored variable

$$z \equiv \frac{\hat{\mathcal{L}}_{\text{IBS}} - \mathcal{L}_{\text{true}}}{\sqrt{\text{Var}[\hat{\mathcal{L}}_{\text{IBS}}]}}, \tag{19}$$

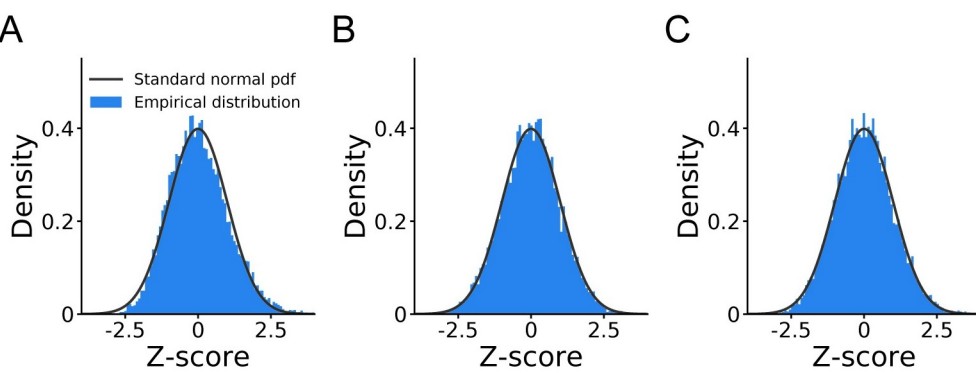

**Fig 3. Normality of the IBS estimator. A.** $z$-score plot for the total number of samples used by IBS. **B.** $z$-score plot for the estimates returned by IBS, using the exact variance formula for known probability. **C.** Calibration plot for the estimates returned by IBS, using the variance estimate from Eq 16. These figures show that the number of samples taken by IBS and the estimated log-likelihood are (approximately) Gaussian, and that the variance estimate from Eq 16 is calibrated.

where here $\mathrm{Var}[\hat{\mathcal{L}}_{\mathrm{IBS}}]$ is the exact variance of the estimator computed via Eq 15. The histogram of $z$-scores in Fig 3B is again very close to a standard normal.

Finally, in practical scenarios we do not know the true likelihoods, so the key question is whether we can obtain valid estimates of the variance of $\hat{\mathcal{L}}_{\mathrm{IBS}}$ via Eq 16. If such an estimate is correctly calibrated, the distribution of $z$-scores should remain approximately Gaussian if we use Eq 16 for the denominator of Eq 19. Indeed, the calibration plot in Fig 3C shows an excellent match with a standard normal, confirming that our proposed estimator of the variance is well calibrated.

## Results

In this section, we examine the performance of IBS and fixed sampling on several realistic model-fitting problems of increasing complexity. The example problems we consider here model tasks drawn from psychophysics and cognitive science: an Orientation discrimination experiment; a Change localization task; and playing a Four-in-a-row game that involves complex sequential decision making. For the first problem, we can derive the exact analytical expression for the log-likelihood; for the second problem, we have an integral expression for the log-likelihood that we can approximate numerically; and finally, for the third problem, we are in the true scenario in which the log-likelihood is intractable.

The rationale for our numerical experiments is that so far we have analyzed fixed sampling and IBS in terms of the bias in their log-likelihood estimates for individual parameter vectors. However, these log-likelihood estimators are often used as elements of a more complex statistical procedure, such as maximum-likelihood estimation. It is plausible that biases in log-likelihood estimates will lead to biases in parameter estimates obtained by maximizing the log-likelihood, but the exact relationship between those biases, and the role of variance in optimization is not immediate. Similarly, it is unclear how bias and variance of individual log-likelihood estimates will affect the estimate of the *maximum* log-likelihood, often used for model selection (e.g., unbiased estimates of the log-likelihood do not guarantee an error-free estimate of the maximum log-likelihood, which is affected by other factors; see later "Log-likelihood loss"). Therefore, we conduct an empirical study showing that, in practice, IBS leads to more accurate parameter and maximum log-likelihood estimates than fixed sampling, given the same budget of computational resources.

First, we describe the procedure used to perform our numerical experiments. Code to run all our numerical experiments and analyses is available at the following link: https://github.com/basvanopheusden/ibs-development.

## Procedure

For each problem, we simulate data from the generative model given different known settings $\boldsymbol{\theta}_{\mathrm{true}}$ of model parameters, and we compare the accuracy (and other statistics) of both IBS and fixed sampling in recovering the true data-generating parameters through maximum-likelihood estimation. Since these methods provide noisy and possibly biased estimates of $\mathcal{L}(\boldsymbol{\theta})$, and due to variability in the simulated datasets, the estimates $\hat{\boldsymbol{\theta}}_{\mathrm{MLE}}$ that result from optimizing the log-likelihood will also be noisy and possibly biased. To explore performance in a variety of settings, and to account for variability in the data-generation process, for each problem we consider $40 \cdot D$ different parameter settings, where $D$ is the number of model parameters (that is, the dimension of $\boldsymbol{\theta}$), and for each parameter setting we generate 100 distinct synthetic datasets.

For each dataset, we compare fixed sampling with different numbers of samples $M$ (from $M = 1$ to $M = 50$ or $M = 100$, depending on the problem), and IBS with different number of 'repeats' $R$, as defined in the Methods (from $R = 1$ to up to $R = 50$, depending on the problem). In each scenario, we directly compare the two methods in terms of number of samples by computing the *average* number of samples used by IBS for a given number of repeats $R$. To prevent IBS from 'hanging' on particularly bad parameter vectors, we adopt the 'early stopping threshold' technique described in S1 Appendix C.1. Finally, if available, we also test the performance of maximum-likelihood estimation using the 'exact' log-likelihood function (calculated either analytically or via numerical integration).

For all methods, we maximize the log-likelihood with Bayesian Adaptive Direct Search (BADS [27]; github.com/lacerbi/bads), a hybrid Bayesian optimization algorithm based on the mesh-adaptive direct search framework [44], which affords a fast, robust exploration of the function landscape via Gaussian process surrogates. Briefly, BADS works by alternating between two stages: in the Poll stage, the algorithm evaluates points in a random mesh surrounding the current point, in a fairly model-free way; in the Search stage, following the principles of Bayesian optimization [33], the algorithm builds a local Gaussian process model of the target function, and chooses the next point by taking into account both mean and variance of the surrogate model, balancing exploration of unknown but promising regions and exploitation of regions known to be high-valued (for maximization). By combining model-free and powerful model-based search, BADS has been shown to be much more effective than alternative optimization methods particularly when dealing with stochastic objective functions, and with a relatively limited budget of a few hundreds to a few thousands function evaluations [27]. We refer the interested reader to [27] and to the extensive online documentation for further information about the algorithm.

## Orientation discrimination

The first task we simulate is an orientation discrimination task, in which a participant observes an oriented patch on a screen, and indicates whether they believe it was rotated leftwards or rightwards with respect to a reference line (see Fig 4A). Here, on each trial the stimulus $s$ is the orientation of the patch with respect to the reference (in degrees), and the response $r$ is 'rightwards' or 'leftwards'.

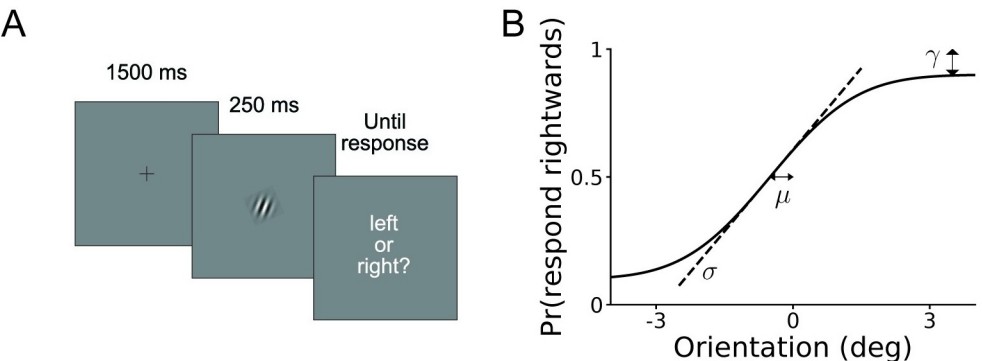

**Fig 4. Orientation discrimination task. A**. Trial structure of the simulated task. A oriented patch appears on a screen for 250 ms, after which participants decide whether it is rotated rightwards or leftwards with respect to a vertical reference. **B**. Graphical illustration of the behavioral model, which specifies the probability of choosing rightwards as a function of the true stimulus orientation. The three model parameters $\sigma$, $\mu$, and $\gamma$ correspond to the (inverse) slope, horizontal offset and (double) asymptote of the psychometric curve, as per Eq 20. Note that we parametrize the model with $\eta \equiv \log \sigma$.

For each dataset, we simulated $N = 600$ trials, drawing on each trial the stimulus $s$ from a Gaussian distribution with mean 0˚ (the reference) and standard deviation 3˚. The generative model assumes that the observer makes a noisy measurement $x$ of the stimulus, which is normally distributed with mean $s$ and standard deviation $\sigma$, as per standard signal detection theory [45]. They then respond 'rightwards' if $x$ is larger than $\mu$ (a parameter which captures response bias, or an incorrect memory of the reference line) and 'leftward' otherwise. However, a fraction of the time, given by the lapse rate $\gamma \in (0, 1]$, the observer guesses randomly. We visually illustrate the model in Fig 4B. For both theoretical reasons and numerical convenience, we parametrize the slope $\sigma$ as $\eta \equiv \log \sigma$. Thus, the model has parameter vector $\boldsymbol{\theta} = (\eta, \mu, \gamma)$.

We can derive the likelihood of each trial analytically:

$$\Pr(\text{'rightwards' response}|s, \boldsymbol{\theta}) = \frac{\gamma}{2} + (1 - \gamma)\Phi\left(\frac{s - \mu}{\sigma}\right), \tag{20}$$

where $\Phi(x)$ is the cumulative normal distribution function. Eq 20 takes the form of a typical psychometric function [46]. Note that in this section we use Gaussian distributions for circularly distributed variables, which is justified under the assumption that both the stimulus distribution and the measurement noise are small. For more details about the numerical experiments, see S1 Appendix B.1.

In Fig 5, we show the parameter recovery using fixed sampling, IBS and the exact log-likelihood function from Eq 20. First, we show that IBS can estimate the sensory noise parameter $\eta$ and lapse rate $\gamma$ more accurately than fixed sampling while using on average the same or fewer samples (Fig 5A and 5D). For visualization purposes, we show here a representative example with $R = 1$ or $R = 3$ repeats of IBS and $M = 10$ or $M = 20$ fixed samples (see Fig 6 in S1 Appendix for the plots with all tested values of $R$ and $M$). As baseline, we also plot the mean and standard deviation of exact maximum-likelihood estimation, which is imperfect due to the finite data size (600 trials), and stochasticity and heuristics used in the optimization algorithm. We omit results for estimates of the response bias $\mu$, since even fixed sampling can match the performance of exact MLE with only 1 sample per trial.

Next, we fix $\eta_{\text{true}} \equiv \log \sigma_{\text{true}} = \log 2˚$, $\mu_{\text{true}} = 0.1˚$, $\gamma_{\text{true}} = 0.1$ and plot the mean and standard deviation of the estimated $\hat{\eta}$ and $\hat{\gamma}$ across 100 simulated data sets as a function of the (average) number of samples per trial used by IBS or fixed sampling (Fig 5B and 5E). We find that fixed sampling is highly sensitive to the number of samples, and with less than 20 samples per trial, its estimate of $\eta$ is strongly biased. Estimating $\gamma$ accurately remains unattainable even with 100 samples per trial. By contrast, IBS estimates $\eta$ and $\gamma$ reasonably accurately regardless of the number of samples per trial. IBS has a slight tendency to underestimate $\gamma$, which is a result of an interaction of the uncertainty handling in BADS with our choice of model parametrization and parameter bounds. In general, estimating lapse rates is notoriously prone to biases [47].

Finally, we measure the root mean squared error (RMSE) of IBS, fixed sampling and the exact solution, averaged across all simulated data sets, as a function of number of samples per trial (Fig 5C and 5F). This analysis confirms the same pattern: fixed sampling makes large errors in estimating $\eta$ with fewer than 20 samples, and for $\gamma$ it requires as many as 100 samples per trial to become approximately unbiased. IBS outperforms fixed sampling for both parameters and any number of samples, and even with as few as 2 or 3 repeats comes close to matching the RMSE of exact maximum-likelihood inference.

## Change localization

The second problem we consider is a typical 'change localization' task (see Fig 6A), in which participants observe a display of 6 oriented patches, and after a short inter-stimulus interval, a second display of 6 patches [48]. Of these patches, 5 are identical between displays and one

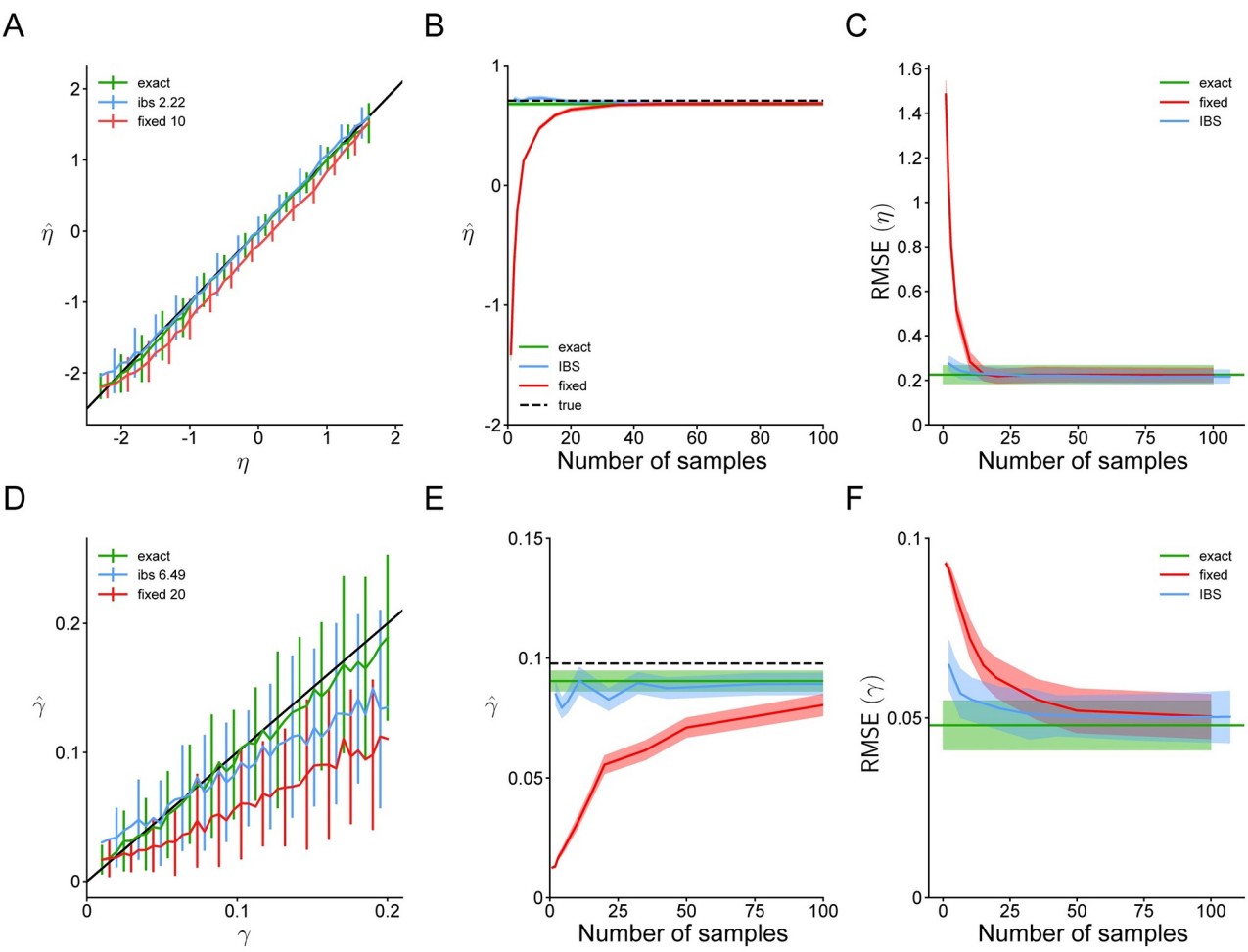

**Fig 5. Parameter recovery for the orientation discrimination model. A.** Estimated values of $\eta \equiv \log \sigma$ as a function of the true $\eta$ in simulated data using IBS with $R = 1$ repeat (blue), fixed sampling with $M = 10$ (red) or the exact likelihood function (green). The black line denotes equality. Error bars indicate standard deviation across 100 simulated data sets. IBS uses on average 2.22 samples per trial. **B.** Mean and standard error (shaded regions) of estimates of $\eta$ for 100 simulated data sets with $\eta_{\text{true}} = \log 2°$, using fixed sampling, IBS or the exact likelihood function. For fixed sampling and IBS, we plot mean and standard error as a function of the (average) number of samples used. **C.** Root mean squared error (RMSE) of estimates of $\eta$, averaged across the range of $\eta_{\text{true}}$ in **A**, as a function of the number of samples used by IBS or fixed sampling. Shaded regions denote ±1 standard deviation across the 100 simulated data sets. We also plot the RMSE of exact maximum-likelihood estimation, which is nonzero since we simulated data sets with only 600 trials. **D-F** Same, for $\gamma$ (with $R = 3$, $M = 20$ in panel **D**). These results demonstrate that IBS estimates parameters of the model for orientation discrimination more accurately than fixed sampling using equally many or even fewer samples.

denoted by $c \in \{1, \ldots, 6\}$ will have changed orientation. The participant responds by indicating which patch they believe changed orientation. Here, on each trial the stimulus $s$ is a vector of 12 elements corresponding to a vector of orientations (in degrees) of the six patches in the first display, concatenated with the vector of orientation of the six patches in the second display. The response $r \in \{1, \ldots, 6\}$ is the patch reported by the participant.

For each dataset, we simulated $N = 400$ trials. On each trial, the patches on the first display are all independently drawn from a uniform distribution Uniform[0, 360]. For the second display, we randomly select one of the patches and change its orientation by an amount drawn from a von Mises distribution centered at 0° with concentration parameter $\kappa_s = 1$. A von Mises distribution is the equivalent of a Gaussian distribution in circular space, and the concentration parameter is monotonically related to the precision (inverse variance) of the

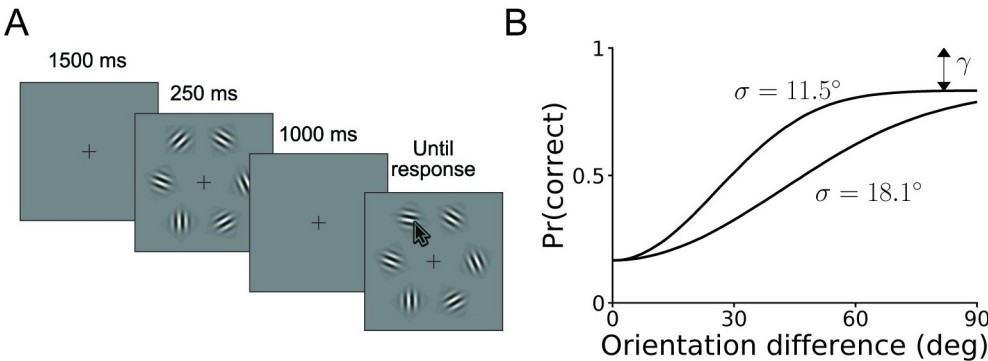

**Fig 6. Change localization task. A**. Trial structure of the simulated task. While the participant fixates on a cross, 6 oriented patches appear for 250 ms, disappear and the re-appear after a delay. In the second display, one patch will have changed orientation, in this example the top left. The participant indicates with a mouse click which patch they believe changed. **B**. The generative model is fully characterized by the proportion correct as function of model parameters and circular distance between the orientations of the changed patch in its first and second presentation (see text). Here we plot this curve for two values of $\eta \equiv \log \sigma$. In both curves, $\gamma = 0.2$. We can read off $\eta$ from the slope and $\gamma$ from the asymptote.

distribution. Note that, for mathematical convenience (but without loss of generality) we assume that patch orientations are defined on the whole circle, whereas in fact they are defined on the half-circle [0˚, 180˚).

The generative model assumes that participants independently measure the orientation of each patch in both displays. For each patch, the measurement distribution is a von Mises centered on the true orientation with concentration parameter $\kappa$, representing sensory precision. The participant then selects the patch for which the absolute circular difference of the measurements between the first and second display is largest. This model too includes a lapse rate $\gamma \in (0, 1]$, the probability with which the participant guesses uniformly randomly across responses.

Since thinking in terms of concentration parameter is not particularly intuitive, we reparametrize participants' sensory noise as $\eta \equiv \log \sigma \equiv -\frac{1}{2} \log \kappa$, since in the limit $\kappa \gg 1$, the von Mises distribution with concentration parameter $\kappa$ tends to a Gaussian distribution with standard deviation $\sigma = \frac{1}{\sqrt{\kappa}}$. The model has then two parameters, $\boldsymbol{\theta} = (\eta, \gamma)$.

We can express the trial likelihood for the change localization model in an integral form that does not have a known analytical solution (see S1 Appendix B.2 for a derivation). We can, however, evaluate the integral numerically, which can take a few seconds for a high-precision likelihood evaluation across all trials in a dataset. The key quantity in the computation of the trial likelihood is $\Delta_s^{(c)}$, the difference in orientation between the changed stimulus at position $c$ between the first and second display. We plot the probability of a correct response, $P_{\text{correct}}(\Delta_s^{(c)}; \boldsymbol{\theta})$, as a function of $\Delta_s^{(c)}$ in Fig 6B. As expected, the probability of a correct response increases monotonically with the amount of change, with the slope being modulated by sensory noise and the asymptote by the lapse rate (but also by the sensory noise, for large noise, as we will discuss later). For more details about the numerical experiments, see S1 Appendix B.2.

In Fig 7, we compare the performance of IBS, fixed sampling and the 'exact' log-likelihood evaluated through numerical integration. As before, IBS estimates both $\eta$ and $\gamma$ more accurately with fewer samples than fixed sampling (Fig 7A and 7D). As an example, we show IBS with $R = 1$ repeats and fixed sampling with $M = 20$ or $M = 50$; the full results with all tested values of $R$ and $M$ are reported in Fig 7 in S1 Appendix.

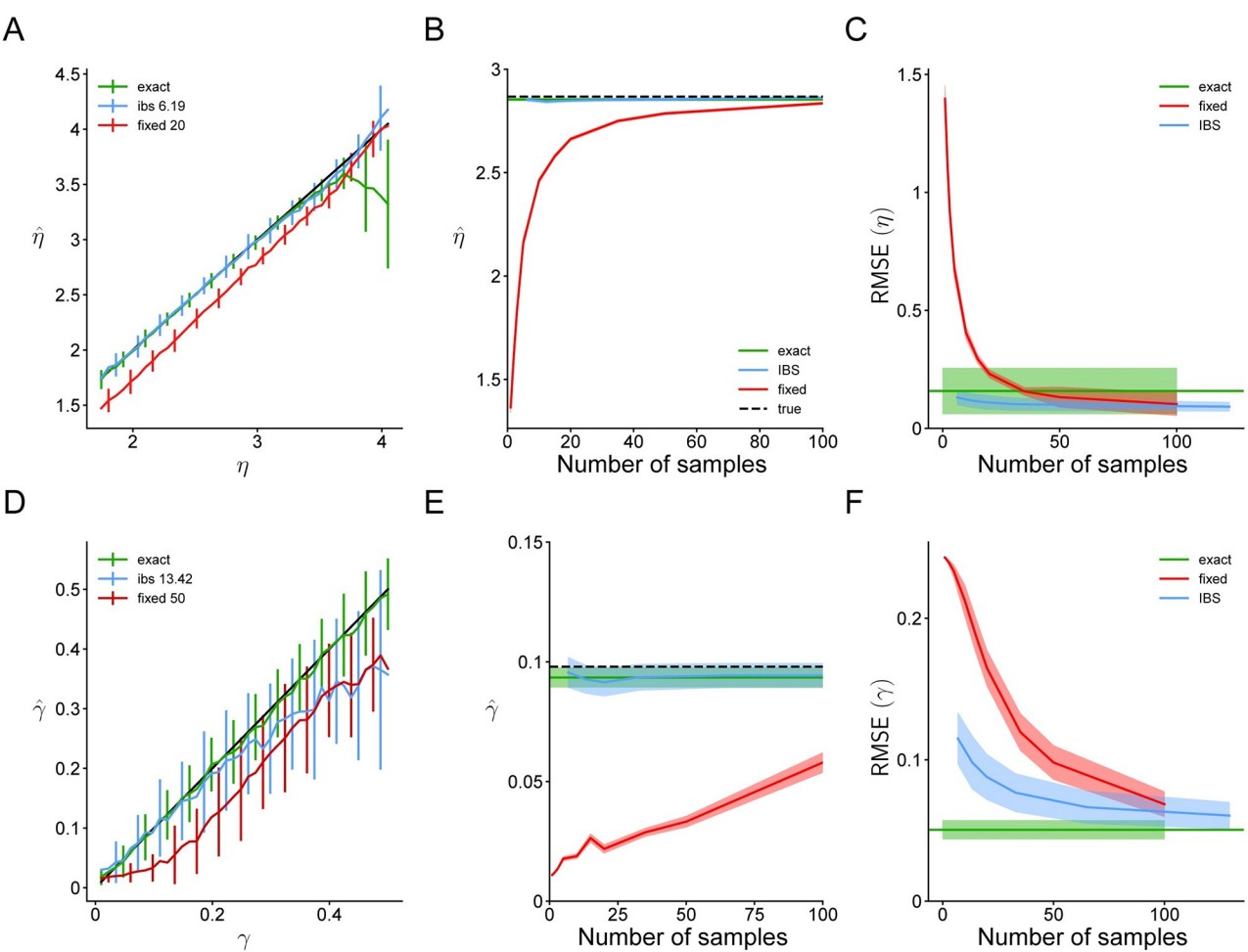

**Fig 7. Parameter recovery for the change localization model.** Same as Fig 5, for the change localization experiment and estimates of $\eta \equiv \log \sigma$ and $\gamma$. In panel **A** we show results for $R = 1$ and $M = 20$; in panel **D**, $R = 2$ and $M = 50$.

Interestingly, maximum-likelihood estimation via the 'exact' method provides biased estimates of $\eta$ when the noise is high. This is because sensory noise and lapse become empirically non-identifiable for large $\eta$, as large noise produces a nearly-flat response distribution, which is indistinguishable from lapse. For these particular settings of $\boldsymbol{\theta}_{\text{true}}$, due to the interaction between noisy log-likelihood evaluations and the optimization method, IBS and fixed sampling perform better at recovering $\eta$ than the 'exact' method, but it does not necessarily hold true in general. Issues of identifiability can be ameliorated by using Bayesian inference instead of maximum-likelihood estimation [49].

In Fig 7B and 7E, we show the estimates of fixed sampling and IBS for simulated data with $\eta_{\text{true}} \equiv \log \sigma_{\text{true}} = \log 17.2°$ and $\gamma_{\text{true}} = 0.1$, and find that fixed sampling substantially underestimates $\eta$ when using less then 50 samples, and underestimates $\gamma$ even with 100 samples per trial. By contrast, IBS produces parameter estimates with relatively little bias and standard deviation close to that of exact maximum-likelihood estimation. Finally, in Fig 7C and 7F we show that IBS has lower RMSE than fixed sampling for both parameters when compared on equal terms of number of samples.

**Fig 8. Four-in-a-row task. A**. Example board configuration in the 4-in-a-row task, in which two players alternate placing pieces (white or black circles) on a 4-by-9 board (gray grid), and the first player to get 4 pieces in a row wins. In this example, the black player can win by placing a piece on the square on the bottom row, third column. **B**. Illustration of features used in the value function of the heuristic search model (Eq 21). For details on the model, see S1 Appendix B.3 and [14].

## Four-in-a-row game

The third problem we examine is a complex sequential decision-making task, a variant of tic-tac-toe in which two players compete to place 4 pieces in a row, column or diagonal on a 4-by-9 board (see Fig 8A). Previous work showed that people's decision-making process in this game can be modeled accurately as *heuristic search* [14]. A heuristic search algorithm makes a move in a given board state by searching through a decision tree of move sequences starting at that board state for a number of moves into the future. To decide which candidate future moves to include in the tree, the algorithm uses a *value function* defined as

$$V(\text{board}, \text{move}) = \sum_{i=1}^{n_f} w_i f_i(\text{board}, \text{move}) + \mathcal{N}(0, \sigma^2), \tag{21}$$

in which $f_i$ denotes a set of $n_f$ features (i.e., configurations of pieces on the board, such as 'three pieces on a row of the same color'; see Fig 8B), $w_i \in \mathbb{R}$ the corresponding feature weights, and $\sigma > 0$ is a model parameter which controls value noise. As before, we parameterize the model with $\eta \equiv \log \sigma$.

The interpretation of this value function is that moves which lead to a high value $V(\text{board}, \text{move})$ are given priority in the search algorithm, and the model is more likely to make those moves. As a heuristic to reduce the search space, any moves for which the value $V(\text{board}, \text{move})$ is less than that of the highest-value move minus a *threshold* parameter $\xi > 0$ are *pruned* from the tree and never considered as viable options. Finally, when evaluating $V(\text{board}, \text{move})$, the model stochastically omits features from the sum $\sum_{i=1}^{n_f} w_i f_i$; the probability for any feature to be omitted (or *dropped*) is independent with probability $\delta \in [0, 1]$ (the *drop rate*). Previous work considered various heuristic search models with different feature sets, and estimated the value of feature weights $w_i$ as well as the size (number of nodes) of the decision tree based on human data [14]. Here, we consider a reduced model in which the feature identity $f_i$, feature weights $w_i$ and tree size are fixed (see S1 Appendix B.3 for their values). Thus, the current model has three parameters, $\boldsymbol{\theta} = (\eta, \xi, \delta)$.

Note that even though the 4-in-a-row task is a sequential game, the heuristic search model makes an independent choice on each move, with the 'stimulus' $s$ on each trial being the current board state. Hence, the model satisfies the conditional independence assumptions of Eqs 2 and 6. Note also that, even though the heuristic search algorithm can be specified as a generative 'simulator' which we can query to make moves in any board position, we have no way of calculating the distribution over its moves, since this would require integrating over all possible trees it could build, features which may be dropped, and realizations of the value noise.

Therefore, we are in the scenario in which log-likelihood estimation is only possible by simulation, and we cannot compare the performance of fixed sampling or IBS to any 'exact' method.

To generate synthetic data sets for the 4-in-a-row task, we first compiled a set of 5482 board positions which occurred in human-versus-human play [14]. For each data set, we then randomly sampled $N = 100$ board positions without replacement which we used as 'stimuli' for each trial, and sampled a move from the heuristic search algorithm for each position to use as 'responses'. For more details about the numerical experiments, see S1 Appendix B.3.

In Fig 9, we perform the same tests as before, comparing fixed sampling and IBS, but lacking any 'exact' estimation method. Due to the high computational complexity of the model, we only consider IBS with up to $R = 3$ repeats, corresponding to $\sim 80$ samples. The full results with all tested values of $R$ and $M$ are reported in Fig 8 in S1 Appendix. As a specific example for Fig 9B, 9E and 9H we show the estimates of fixed sampling and IBS for simulated data with $\eta_{\text{true}} \equiv \log \sigma_{\text{true}} = \log 1$, pruning threshold $\xi_{\text{true}} = 5$ and $\delta_{\text{true}} = 0.2$.

Fixed sampling underestimates the value noise $\eta$, even when using $M = 100$ samples, whereas IBS estimates it accurately with 4 times fewer samples (Fig 9A). This bias of fixed sampling gets worse with fewer samples (Fig 9B), and overall, IBS outperforms fixed sampling when compared on equal terms (Fig 9C). The same holds true for the pruning threshold $\xi$. IBS estimates $\xi$ about equally well as fixed sampling, but with about half as many samples (Fig 9D), fixed sampling is severely biased when using too few samples (Fig 9E) and overall, IBS outperforms fixed sampling.

The results are slightly more complicated for the feature drop rate $\delta$. As before, fixed sampling produces strongly biased estimates of $\delta$ with up to 35 samples (Fig 9G), and the bias increases when using fewer samples (Fig 9H). However, for this parameter IBS is also biased, but towards 0.25 (Fig 9G and 9H), which is the midpoint of the 'plausible' upper and lower bounds used as reference by the optimization algorithm (see S1 Appendix B.3 for details). This bias can be interpreted as a form of regression towards the mean; likely a by-product of the optimization algorithm struggling with a low signal-to-noise ratio for this parameter and these settings (i.e., a nearly flat likelihood landscape for the amount of estimation noise on the log-likelihood). The negative bias of fixed sampling helps to reduce its variance in the low-$\delta$ regime, and therefore in terms of RMSE, fixed sampling performs similarly to IBS for this parameter (Fig 9I).

## Log-likelihood loss

In the previous sections, we have analyzed the bias and error of different estimation methods when recovering the generating model parameters in various scenarios. Another important question, crucial for model selection, is how well different methods are able to recover the true maximum log-likelihood. The ability to recover the true parameters and the true maximum log-likelihood are related but distinct properties because, for example, a relatively flat likelihood landscape could yield parameter estimates very far from ground truth, but still afford recovery of a value of the log-likelihood close to the true maximum. We recall that differences in log-likelihood much greater than one point are worrisome as they might significantly affect the outcomes of a model comparison [24–26].

To compute the *log-likelihood loss* of a method for a given data set, we estimate the difference between the 'exact' log-likelihood evaluated at the 'true' maximum-likelihood solution (as found after multiple optimization runs) and the 'exact' log-likelihood of the solution returned by the multi-start optimization procedure for a given method, as described in the "Procedure" section. In terms of methods, we consider IBS and fixed-sampling with different amounts of samples. We perform the analysis for the two scenarios, orientation discrimination

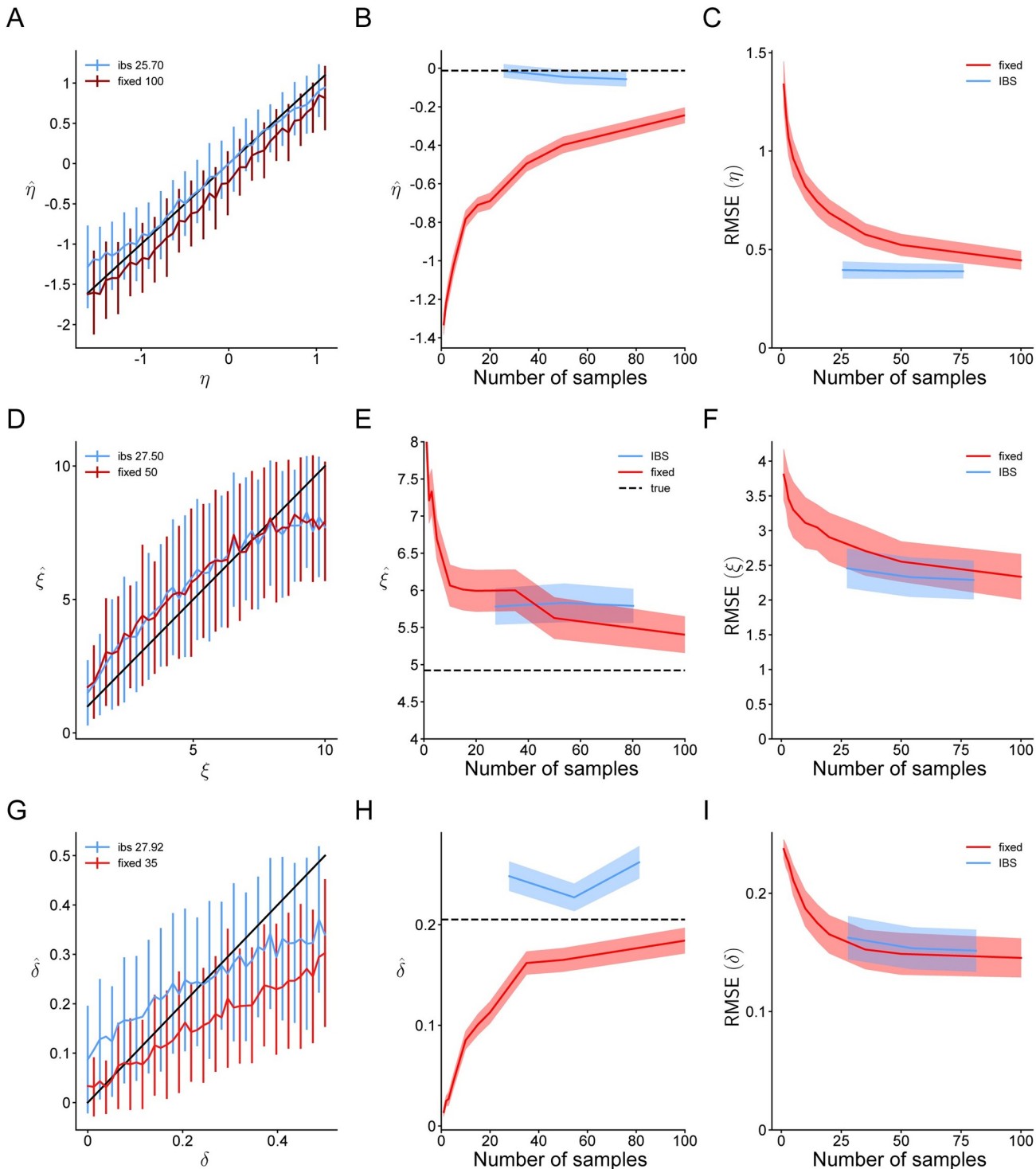

**Fig 9. Parameter recovery for the four-in-a-row model.** Same as Figs 5 and 7, for the 4-in-a-row experiment and estimates of the value noise $\eta \equiv \log \sigma$, pruning threshold $\xi$ and feature drop rate $\delta$. In panel **A** we show results for $R = 1$ and $M = 100$; in panel **D**, $R = 1$ and $M = 50$; in panel **G**, $R = 1$ and $M = 35$.

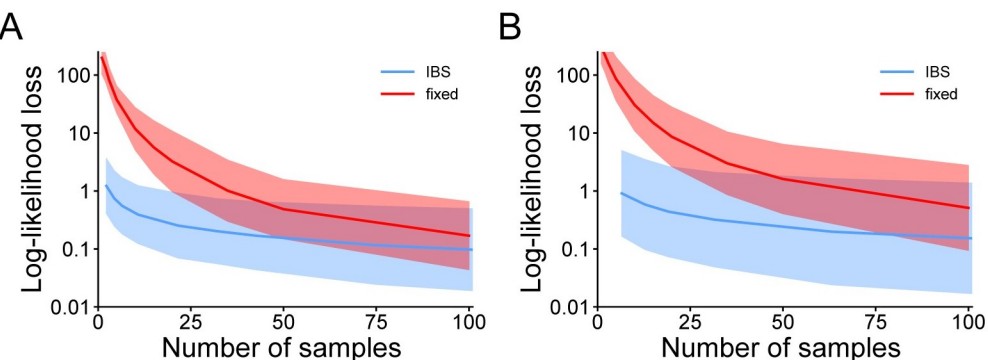

**Fig 10. Log-likelihood loss. A**. Log-likelihood loss with respect to ground truth, as a function of number of samples, for the orientation discrimination task. Lines are mean ±1 standard deviation (in log space) across 120 generating parameter values, with 100 simulated datasets each. **B**. Log-likelihood loss for the change localization task (80 generating parameter values).

and change localization, for which we have access to the exact likelihood, either analytically or numerically.

The results in Fig 10 show that IBS, even with only a few repeats, is able to return solutions which are very close to the true maximum-likelihood solution in terms of log-likelihood (within 1-2 points); whereas fixed sampling remains strongly biased (log-likelihood loss $\gg 1$) even with large number of samples, being thus at risk of inducing wrong inferences in model selection. Note that our analyses of the loss are based on the 'exact' log-likelihood values evaluated at the solution returned by the optimization procedure. In practice, we would not have access to the 'exact' log-likelihood at the solution; but its value can be estimated up to the desired precision with IBS, by taking multiple repeats at the returned solution.

Finally, the results in Fig 10 also display clearly that, while IBS is unbiased in estimating the log-likelihood *for a given parameter setting θ*, the maximum-likelihood procedure per se will have some error. Due to estimation noise and specific features of the data, model, and stochastic optimization method at hand, the returned solution will rarely be the true maximum-likelihood solution, and thus, by definition, the value of the log-likelihood at the solution will *underestimate* the true value of the maximum log-likelihood. Still, Fig 10 shows that the underestimation error, at least in the IBS case, tends to be acceptable, as opposed to the large errors obtained with fixed sampling.

## Summary

The results in this section demonstrate that in realistic scenarios, fixed sampling with too few samples causes substantial biases in parameter and maximum log-likelihood estimates, whereas inverse binomial sampling is much more accurate and robust to the number of samples used. Across all 3 models and all parameters, IBS yields parameter estimates with little bias and RMSE, close to that of 'exact' maximum-likelihood estimation, even when using only a handful of repeats ($R$ between 1 and 5). Conversely, fixed sampling yields substantially biased parameter estimates when using too few samples per trial, especially for parameters which control decision noise, such as measurement noise and lapse rates in the two perceptual decision-making tasks, and value noise in the 4-in-a-row task. Moreover, for the two models for which we have access to 'exact' log-likelihood estimates, we found that IBS is able to recover maximum-likelihood solutions close to the true maximum log-likelihood, whereas fixed sampling remains severely biased even for many samples.

It is true that, given a *large enough* number of samples, fixed sampling is eventually able to recover most parameters and maximum log-likelihood values with reasonable accuracy. However, we have seen empirically that the number of samples required for reliable estimation varies between tasks, models and parameters of interests. For tasks and models where an exact likelihood or a numerical approximation thereof is unavailable, such as the four-in-a-row problem we examined in the Results, this limitation renders fixed sampling hardly usable in practice. By contrast, IBS automatically chooses the number of samples to allocate to the problem.

Finally, for complex models with a large response space, accurate parameter estimation with fixed sampling will require many more samples per trial than are feasible given the computational time needed to generate them. Therefore, in such scenarios accurate and efficient parameter estimation is only possible with IBS.

## Discussion

In this work, we presented inverse binomial sampling (IBS), a method for estimating the log-likelihood of simulation-based models given an experimental data set. We demonstrated that estimates from IBS are uniformly unbiased, their variance is uniformly bounded, and we introduced a calibrated estimator of the variance. IBS is sample-efficient and, for the purpose of maximum-likelihood estimation, combines naturally with gradient-free optimization algorithms that handle stochastic objective functions, such as Bayesian Adaptive Direct Search (BADS [27]). We compared IBS to fixed sampling and showed that the bias inherent in fixed sampling can cause researchers to draw false conclusions when performing model selection. Moreover, we showed in three realistic scenarios of increasing complexity that maximum-likelihood estimation of model parameters is more accurate with IBS than with fixed sampling with the same average number of samples.

In the rest of this section, we discuss additional applications of IBS, possible extensions, and give some practical usage recommendations.

### Additional applications

We developed inverse binomial sampling for log-likelihood estimation of models with intractable likelihoods, for the purpose of model comparison or fitting model parameters with maximum-likelihood estimation, but IBS has other practical uses.

**Checking analytical or numerical log-likelihood calculations.** We presented IBS as a solution for when the log-likelihood is intractable to compute analytically or numerically. However, even for models where the log-likelihood could be specified, deriving it can be quite involved and time-consuming, and mistakes in the calculation or implementation of the resulting equations are not uncommon. In this scenario, IBS can be useful for:

- quickly prototyping (testing) of new models, as writing the generative model and fitting it to the data is usually much quicker than deriving and implementing the exact log-likelihood;

- checking for derivation or implementation mistakes, as one can compare the *supposedly* 'exact' log-likelihood against estimates from IBS (on real or simulated data);

- assessing the quality of numerical approximations used to calculate the log-likelihood, for example when using methods such as adaptive quadrature for numerical integration [50].

**Estimating entropy and other information-theoretic quantities.** We can also use inverse binomial sampling to estimate the entropy of an arbitrary discrete probability

distribution $\Pr(x)$, with $x \in \Omega$, a discrete set (see [51] for an introduction to information theory). To do this, we first draw a sample $x$ from the distribution, then use IBS to estimate log $\Pr(x)$. The first sample and the samples in IBS are independent, and therefore we can calculate the expected value of the outcome of IBS,

$$\mathbb{E}[\hat{\mathcal{L}}_{\text{IBS}}] = \mathbb{E}_{x \sim \Pr(\cdot)}[\log \Pr(x)] = \sum_{x \in \Omega} \Pr(x) \ \log \ \Pr(x), \qquad (22)$$

which is the definition of the negative entropy of $\Pr(x)$.

We can use this technique to estimate the entropy of the predicted response distribution of a generative model with a given parameter vector on any trial. For example, such quantity could be used in a behavioral model to test for the generalized Hick-Hyman law, that states that reaction time is proportional to the entropy of the available choices [52]. Moreover, we can generalize the method to estimate the cross-entropy between two distributions (sample from one, estimate log-likelihood with the other), or the Kullback-Leibler divergence between distributions. We note that all the estimates of these quantities are also uniformly unbiased. Incidentally, the lack of bias in entropy estimates by IBS may be surprising in light of a theorem stating that uniformly unbiased estimators of the entropy given a finite set of samples cannot exist [53]. This theorem does not apply to IBS since its sample size is a stochastic variable. It does, however, prove that one cannot estimate entropy (or similar information-theoretic quantities) with fixed sampling.

## Bayesian inference

In this paper we focused on maximum-likelihood estimation, but another common approach to parameter estimation is Bayesian inference [2]. Bayesian inference has the goal of computing the *posterior distribution* of the parameters given the observations, computed as

$$p(\boldsymbol{\theta}|\mathcal{D}) = \frac{\Pr(\mathcal{D}|\boldsymbol{\theta})p(\boldsymbol{\theta})}{\mathcal{Z}} \qquad \text{with } \mathcal{Z} \equiv \int \Pr(\mathcal{D}|\boldsymbol{\theta})p(\boldsymbol{\theta})d\boldsymbol{\theta}, \qquad (23)$$

where $\Pr(\mathcal{D}|\boldsymbol{\theta})$ is the likelihood, $p(\boldsymbol{\theta})$ the prior density of the parameters (typically assumed continuous), and $\mathcal{Z}$ the normalization constant, known as the *evidence* or *marginal likelihood*, a quantity used for Bayesian model selection due to a number of desirable properties [7]. Since $\mathcal{Z}$ is often hard to compute, many (approximate) Bayesian inference techniques are able to calculate the posterior distribution by having access only to the *unnormalized* posterior, or joint distribution $\Pr(\mathcal{D}|\boldsymbol{\theta})p(\boldsymbol{\theta})$; or equivalently to the log joint $\mathcal{L}(\boldsymbol{\theta}) + \log \ p(\boldsymbol{\theta})$. We see then that IBS could be used to perform Bayesian inference of likelihood-free models by providing a means to compute the log-likelihood in the log joint distribution (the prior is assumed to be a simple distribution which we can express in closed form).

In S1 Appendix C.3, we describe how several approaches to approximate Bayesian inference could be used in conjunction with the unbiased log-likelihood estimates provided by IBS: Markov Chain Monte Carlo [54, 55]; variational inference [56, 57]; and Gaussian process surrogate methods [37, 40], including Variational Bayesian Monte Carlo (VBMC [38, 58]). In particular, [39] demonstrates the effectiveness of IBS, combined with VBMC, for robust and sample-efficient Bayesian inference, using a variety of models from computational and cognitive neuroscience.

Finally, note that the techniques in this paper can be easily applied to maximum-a-posteriori (MAP) estimation—which is not quite Bayesian inference, but more like a regularized form of maximum-likelihood, that still yields a point estimate instead of a full posterior

distribution. MAP estimation is attained by simply adding the log-prior to the log-likelihood in the optimization objective, where the log-prior acts as a regularization term.

## Approximate IBS for continuous responses

So far, we have assumed that the space of possible responses is discrete. This assumption is necessary since, for continuous responses, the probability that a sample from the generative model exactly matches an observed response is zero (technically, *near*-zero since any computer implementation of a real number is finite). For this reason, IBS will never terminate, or at least not within a physically sensible time scale.

A simple approach to make continuous responses discrete is via binning the response space. Alternatively, we recommend an approach inspired by Approximate Bayesian Computation (ABC [15]), which we call Approximate IBS (AIBS). Given a metric $D(\cdot, \cdot)$ to measure distance in response space, and a tolerance threshold $\varepsilon > 0$, we can use IBS to estimate

$$\mathcal{L}_\varepsilon(\boldsymbol{\theta}) = \sum_{i=1}^{N} \log \frac{\Pr(D(\tilde{\boldsymbol{r}}_i, \boldsymbol{r}_i) \leq \varepsilon | \boldsymbol{s}_i, \boldsymbol{\theta})}{|B_\varepsilon(\boldsymbol{r}_i)|}, \tag{24}$$

where the $\tilde{\boldsymbol{r}}_i$ are responses drawn from the generative model, and $|B_\varepsilon(\boldsymbol{r}_i)|$ denotes the volume of the set of responses whose distance from $\boldsymbol{r}_i$ is no more than $\varepsilon$.

The $\varepsilon$-approximate log-likelihood in Eq 24 can then be used as normal for maximum-likelihood estimation or Bayesian inference. As $\varepsilon \to 0$, the approximate likelihood tends to the true likelihood, under some regularity conditions which we leave to explore for future work (see [59] for a similar proof for ABC). However, the expected number of samples used by IBS diverges in that limit, so in practice there is a lower bound for $\varepsilon$ that is feasible and one needs to extrapolate to the $\varepsilon = 0$ limit, or be satisfied to perform inference with an $\varepsilon$-approximate likelihood.

The common idea between AIBS and ABC is that they both use a distance metric to judge similarity between simulated samples and data. However, ABC commonly bases the comparison on *summary statistics* of the data (which may not be *sufficient* statistics, and thus not capture all aspects of the data); whereas AIBS uses the full responses. Secondly, ABC in practice requires dedicated algorithms to perform parameter estimation and inference (basic techniques, such as rejection sampling, can be extremely inefficient); whereas AIBS simply provides a (noisy) log-likelihood, which can then be used in combination with a wider class of likelihood-based inference methods, as long as they support noisy estimates (see S1 Appendix C.3 for some examples). We leave a further analysis of AIBS, and a comparison with other likelihood-free inference approaches, as a promising direction for future work.

## Usage recommendations

We conclude with a number of recommendations for researchers who want to fit a model to a data set, having access only to a simulator or generative model.

- First, try to derive a closed-form analytic expression for the log-likelihood of the model. If this is tractable, validate that the log-likelihood is free of implementation mistakes by comparing its output against log-likelihood estimates obtained by IBS with well-chosen test trials and model parameters.

- If exact analytics are intractable, find an analytical or numerical approximation, for example using variational inference or Riemannian integration, and once again validate the quality of the approximation using IBS.

- If the model is too complex for analytical or numerical approximations, estimate the log-likelihood using inverse binomial sampling.

- Finally, perform inference using the analytical, numerical, or IBS-based log-likelihood function with a sample-efficient inference algorithm, such as those based on Gaussian process surrogate modeling. For maximum-likelihood (or maximum-a-posteriori) estimation, hybrid Bayesian optimization methods have proved to be quite effective [27].

**Avoiding infinite loops.** One issue of IBS is that it can 'hang', in the sense that the implementation of the estimator can run indefinitely, without returning an answer, if the simulator is unable to match a particularly unlikely observation. This is a natural behavior of IBS that stems from its efficiency in allocating samples, as we examined in the "Computational time" section. We recommend two easy solutions to avoid infinite loops:

- Implement a 'lapse rate' $\gamma \in (0, 1)$ in the simulator model, which represents the probability of a completely random response (typically uniform across all possible responses). The lapse rate could be fixed to a small, non-zero value (e.g., $\gamma = 0.01$), or let as a free model parameter; in which case, ensure that the *minimum* lapse rate is a small, non-zero value (e.g., $\gamma_{min} = 0.005$).

- Introduce an early-stopping threshold, such that IBS stops sampling when the estimated log-likelihood of the entire data set goes below a threshold $\mathcal{L}_{lower}$ (see S1 Appendix C.1).

We implemented both of these solutions in our analyses in the Results.

## Supporting information

**S1 Appendix. Supplemental methods. A**. Further theoretical analyses: Why inverse binomial sampling works; Analysis of bias of fixed sampling; Derivation of IBS variance; Estimator variance and information inequality; A Bayesian derivation of the IBS estimator; Estimator RMSE. **B**. Experimental details: Orientation discrimination; Change localization; Four-in-a-row game. **C**. Improvements of IBS and further applications: Early stopping threshold; Reducing variance by trial-dependent repeated sampling; Bayesian inference with IBS. (PDF)

## Acknowledgments

This work has utilized the NYU IT High Performance Computing resources and services. We thank Aspen Yoo for help with Figs 4 and 6 and useful comments on the manuscript, and Michael Landy for helpful discussion about the derivation of the variance of the IBS estimator. Luigi Acerbi thanks the Academy of Finland Flagship programme: Finnish Center for Artificial Intelligence (FCAI).

## Author Contributions

**Conceptualization:** Bas van Opheusden, Luigi Acerbi, Wei Ji Ma.

**Data curation:** Bas van Opheusden.

**Formal analysis:** Bas van Opheusden, Luigi Acerbi.

**Funding acquisition:** Wei Ji Ma.

**Investigation:** Bas van Opheusden, Luigi Acerbi.

**Methodology:** Bas van Opheusden, Luigi Acerbi.

**Software:** Bas van Opheusden, Luigi Acerbi.

**Supervision:** Bas van Opheusden, Luigi Acerbi, Wei Ji Ma.

**Validation:** Bas van Opheusden, Luigi Acerbi.

**Visualization:** Bas van Opheusden, Luigi Acerbi.

**Writing – original draft:** Bas van Opheusden, Luigi Acerbi.

**Writing – review & editing:** Bas van Opheusden, Luigi Acerbi, Wei Ji Ma.

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
