## [Decision Letter · Decision Letter 0]

5 May 2020

Dear Dr. van Opheusden,

Thank you very much for submitting your manuscript "Unbiased and Efficient Log-Likelihood Estimation with Inverse Binomial Sampling" for consideration at PLOS Computational Biology.

As with all papers reviewed by the journal, your manuscript was reviewed by members of the editorial board and by several independent reviewers.

Your paper was in generally well received, but some major issues need to be addressed.

You will find them in details in the reviews, but mainly:

- the paper is very long and appears difficult to read. We certainly don't want to cut methodological details out (even more so being this a methods paper), but you can think of a more agile organization

- the novelty and generalizability of the model both need to be further stressed, in particular where, when, and how someone should resort to this method

In light of the reviews (below this email), we would like to invite the resubmission of a significantly-revised version that takes into account the reviewers' comments.

We cannot make any decision about publication until we have seen the revised manuscript and your response to the reviewers' comments. Your revised manuscript is also likely to be sent to reviewers for further evaluation.

Sincerely,

Daniele Marinazzo

Deputy Editor

PLOS Computational Biology

Reviewer's Responses to Questions

**Comments to the Authors:**

Reviewer #1: REVIEW OF UNBIASED AND EFFICIENT LOG-LIKELIHOOD ESTIMATION WITH INVERSE BINOMIAL SAMPLING

Summary

This paper introduces Inverse Binomial Sampling (IBS) as a method to

compute (log) likelihoods when exact computation is intractable. It is

shown that IBS provides unbiased estimates of the likelihood and that it

compares favourably to an alternative fixed sampling procedure.

Evaluation

I think the IBS procedure is an elegant method to estimate the log

likelihood from simulation when analytical computation is infeasible. I

have not come across this method explicitly, and I think it deserves to

be brought to the attention of researchers. I do find the paper rather

long however, and I think the method could be presented and evaluated in

a manuscript almost half as long. In addition, I think the paper could be

clearer on a number of points, detailed below.

Major issues

1. I found the paper overly long. As I see it, the main message is (1)

IBS offers a method to obtain an unbiased estimate of

$\\log L = \\sum_{i=1}^n \\log p_i$ by simulation from a generative

model. It also offers a way to match simulation cost to the

difficulty of estimating small p_(i). As log p is not a polynomial

function, and “Kolmogorov (1950) showed that the only functions

f (p) for which the fixed sampling policy with M samples allows an

unbiased estimator are polynomials”, there is no unbiased estimate

of log p under fixed sampling. (2) Focusing on the log likelihood of

all data is useful as it allows the central limit theorem to

apply. (3) While IBS is unbiased, its variance is higher than that

of fixed sampling. This may be a situation where minimum bias is

more important than minimum variance. (4) Numerical examples show

that the method performs well and is efficient. I don’t think the

numerical examples need to much explanation of the underlying

experiments or models, and as such the whole paper could probably

take about 50% of the current word count and be more focused and

more impactful. As much of the theoretical developments are derived

as a special case of the more general results in de Groot (1959),

the reader could be referred to there for some of the more intricate

mathematical details.

2. While the fixed sampling scheme has relatively large problems

estimating small probabilities, isn’t the opposite true for IBS.

Does this method not find it more difficult to distinguish between p

= .9 and p = .8 than e.g. p = .05 and p = .15? If so, what does that

imply for model selection between two models which are both quite

good (e.g. nested models with a small change).

3. I think it would be good to more explicitly state that the fixed

sampling procedure provides a biased estimate of log p, but an

unbiased (and minimum variance) estimator exists (and is simple) for

_p_. There are good reasons to focus on the estimation of log p, but

these can be stated more strongly (e.g., even if you have unbiased

estimates of each p_(i), will you then have an unbiased estimate of

∏_(i)p_(i)?

4. I did not find Section 4.5 “Bias or variance?” too convincing.

Estimation typically involves a bias-variance tradeoff. There are

often situations where introducing a little bias is outweighed by

the associated reduction in variance. Rather than plotting bias and

in Fig 1 and 2, it would be more useful to plot something like the

MSE, which incorporates both bias and variance, and discuss the

tradeoff in more detail. Both are important, and I think a statement

like “Bias can cause researchers to confidently draw false

conclusions, variance causes decreased statistical power and lack of

confidence.” is quite misleading. High variance of estimates can

equally lead to false conclusions, if the variance is not properly

taken into account in the uncertainty of the results.

5. The estimate of the variance of the IBS estimator is derived from a

Bayesian procedure. More specifically, “Crucially, Equation 29 shows

that we can recover the IBS estimator as the posterior mean of log q

given K, under the Haldane prior for q. This interpretation allows

us to also define naturally the variance of our estimate for a given

K, as the variance of the posterior over log q,” I don’t see how the

(frequentist) variance of an estimator (i.e. the variance of the

sampling distribution of the estimates produced by an estimator)

equates to the variance of a posterior distribution of a quantity,

for which the posterior mean equals the estimator. Especially when

this is done under a specific Beta(0,0) prior… I may be missing

something, in which case the correspondence could be explained in

more detail.

Minor issues

- “Inverse binomial” is more commonly known as “negative binomial”;

this might also result in a more favourable acronym (IBS is a.k.a.

irritable bowel syndrome)

- p8. “can be a scalar of vector” -> “can be a scalar or vector”

- Equation 1. is defined as the set of all responses and stimuli.

The definition of p(|θ) in Equation 1 contains the stimuli only on

the rhs of the conditional sign, i.e. it is for p(r|s, θ) rather

than p(r, s|θ). These are not equivalent.

- p11. “For example, models with latent dynamics might be able to

generate a full sequence of responses given the stimuli, but it

might not be easy (or even possible) to generate the response in a

given trial, conditional on a specific sequence of previous

responses.” Would it be possible to specify an equivalent version of

IBS counts how long it takes to sample a whole sequence of responses

exactly? This would no doubt take quite some time, but wouldn’t this

also be an unbiased way to estimate log p?

- p13. “in which case the central limit theorem guarantees”. There are

some conditions for the CLT to hold, such as finite variance. It

would be good to check and state that these hold.

- p14. IBS is introduced as a sampling scheme which stops as soon as a

single “hit” is observed. But it can also be defined for any number

of hits greater than 1, Would other forms be less (or maybe more)

useful?

- Equation 11 uses an additional constant = 1 to avoid a negative

infinite log. But might other choices e.g. .5, or .1, not provide

better results?

- p22. “We saw earlier that for fixed-sampling estimators to be

approximately unbiased, one needs at least 1/p_(i)samples, and IBS

does exactly that.” Surely, that only holds in expectation.

- p23. “We note that fixed sampling eventually saturates the

information inequality”. This is better than IBS achieves. But more

importantly, when does it achieve the equivalent of IBS (e.g. a

close 30% to optimum)?

- p23. An estimator for the variance of the estimator of IBS is

derived, but how good is this estimator? Is it unbiased, efficient?

- p25. “fixed sampling can have slightly lower variance.” How is

“slightly” defined. Personally, it looks to me that it can achieve

substantially less variance.

- p25. “Basic rules of probability imply…” Not sure what these “basic”

rules are.. I assume those applying to expected values of sums of

independent variables?

- p26. “By contrast, no optimization algorithm can handle bias.”. This

might be true, but it might also not matter if the bias doesn’t

affect where the maximum is.

- Figure 3A. This distribution looks clearly skewed, so I don’t agree

with the statement “By comparing the histogram of z-scores with a

standard normal in Figure 3A, we see that the total number of

samples is approximately normal.”

- p29. “If such estimate is correctly calibrated,” -> “If such an

estimate is correctly calibrated,”

- p33. “regardless of of the number” -> “regardless of the number”

- Figure 5. “For fixed sampling and IBS, we plot mean and standard

error as a function of the (average) number of samples used.” But

doesn’t the number of samples for IBS depend on p? I don’t think

there is a clearly interpretable comparison between the expected IBS

samples and the M samples for fixed sampling.

- “Interestingly, maximum-likelihood estimation via the ‘exact’ method

provides biased estimates of η when the noise is high.” While it

makes sense that “sensory noise and lapse become empirically

non-identifiable for large η”, why doesn’t this hold for sampling

approximations?

- p44. “The results in Figure 10 show that IBS, even with only a few

repeats, is able” How many repeats?

- Figure 10. “Lines are mean and standard error” Personally, I would

find it more informative to depict quantiles of the distribution of

estimates.

- “whereas fixed sampling remains severely biased even with large

number of samples”. What it “severe” here? In general, such

hyperboles should be avoided (e.g. also on p46 “severely” and “close

to useless”)

- p47. “We developed inverse binomial sampling for log-likelihood

estimation of likelihood-free models”. That seems contradictory. How

can you estimate the log likelihood when there is no likelihood?

Models where the likelihood is infeasible to compute would seem more

accurate.

- p56. “First, we can use the inverse binomial sampling policy to

estimate any analytic function of p.” Why?

- p59. “Therefore, the standard deviation in IBS is close (within 30%)

to its theoretical minimum.” I don’t see how this follows from the

preceding.

- p64. “To avoid wasting computations on ‘bad’ parameter settings, for

IBS we used the ‘early stopping threshold’ technique described in

Section C.1” Doesn’t this reduce the effective number of samples for

the IBS, in which case this should be clearly specified in the

comparisons of samples between IBS and fixed sampling.

Reviewer #2: This paper presents an inverse binomial sampling method for estimating the log-likelihood of simulation-based models given an experimental data set. The authors show that estimators from IBS are uniformly unbiased, their variance is uniformly bounded, and a calibrated estimator for the variance is given. But I have the following comments.

1. When there is not a likelihood function, a nonparametric method such as empirical likelihood has been developed to estimate the probability p_i under the restriction and it is easily implemented. For the current method, the authors did not consider the restriction among p_i. Generally, should consider it. In particular, for fixed method, one can regard the considered case as a multinomial distribution. How is the result? On the other hand, we may not be care the bias rather than variance, for example, the biased estimator.

2. Page 18, section 3.1, line 4: choosing M requires knowledge of p_i on each trial. However, the authors' main purpose is to estimate p_i. Why?

3. Algorithm 1: Generally, theta is unknown. How to sample from model (M,theta,..). Also, how to estimate theta in the considered model.

4. Eq. (16): Is it the variance or the estimated variance of a specific IBS estimator?

5. Section 5.1: what is theta? How to estimate it?

6. Page 33, the second paragraph: It is unclear how to estimate eta and gamma.

7. Page 35, the second paragraph: when the patches are from different distributions, how is the result? In particular, when there are outliers in the considered dataset, how is the result? The proposed method is robust. Also, when the data are from the heavy-tailed distribution, how is the result?

Reviewer #3: The current MS focuses on a common situation where complex computational models are studied based on maximal likelihood (ML) estimation but the likelihood function is unavailable in analytical form or even in a handy numerical form. In such cases one must resort to simulations for estimating the likelihood. The authors show that a commonly used method that estimates likelihood based on a “fixed sample size” simulations has important limitation. For example, it can lead to wrong scientific conclusions in the form of false model comparison inferences and can result in severe biases in estimated parameters. The authors argue the main reason for this is that this method provides biased estimates of log-likelihoods. As an alternative, they present a method based on “inverse binomial sampling” (IBS) that returns non biased log-likelihood estimates. The authors show that IBS has several desirable statistical properties and that in 3 studied examples, it performs better (in parameter-estimation) than the “fixed” sampling method.

I think this is a timely, topical and excellent study that will be relevant for a wide community of modellers across diverse scientific domains and fields. The authors highlight an unfamiliar (or perhaps relatively ignored) method of ML estimation in complex situation, which can improve the way that the community performs model comparisons and parameter estimation. The authors provide a both theoretical arguments and illuminating simulation studies to convince that IBS has several important advantages over the commonly used fixed-sample size method. However, I also have a few concerns that I think must be addressed before the paper can be published. I think that in some cases the connections between the theoretical arguments and the simulations are not sufficiently established and clear. Below, some thoughts and suggestions for improvement.

1) The main argument in sections 2-4 of the paper is that whereas the “fixed” method provides biased estimates of likelihood the IBS provides unbiased estimates. Critically, however, while this is true for any given parameters set it is important to explain to readers that the IBS method will not recover the “true” maximal likelihood value. Suppose that we find the best fitting parameters for some model based on IBS, executing algorithm 1 and using some search (or optimiser) over the parameter-space. Suppose that we can then query an oracle about the true log-likelihood for these best-fitting parameters (i.e., re-calculate the likelihood with infinite precision). Because the parameter-set we found are not the true ML parameters then the likelihood we will obtain must be smaller than the true ML value! Additionally, if we simply rely on the maximal log-likelihood value that was obtained while the search in the parameters space was executed, then we also get biased estimates of ML (because the log likelihood estimate is conditional on being better that other estimates for other parameter-sets; I think it is important to explain this to readers). I think it is critical that the authors discuss these issues as I am worried that some readers may have a false impression that because IBS provides unbiased estimates of the log-likelihood function it can also calculate accurately the true maximal log-likelihood value (when in fact it will necessarily underestimate this value).

Because IBS cannot provide an accurate value of the ML it can also lead to false conclusions (e.g., in model comparison based on BIC where the maximal log-likelihood is plugged). For such uses, the issue of ultimate importance when comparing between the fixed method and IBS, is which method estimates more accurately the maximal log-likelihood value and not whether likelihood is estimated with or without bias for each parameter-set.

It is important to clarify these issues. These issues are somewhat implied in section 5.5 but I think this is too late in the paper and that even in this section, these issue is not discussed explicitly enough.

2) Relatedly, in section 5 the authors compare the methods (fixed vs. IBS) in their ability to recover model-parameters. I believe, there should be some preparatory discussion about the relationship between concepts of accuracy of parameter estimation (sec 5) and concepts that pertain to accuracy in likelihood estimation that were introduced in the earlier sections. How are biases (and variances) in likelihood calculation related to biases (and variances) of estimators of model parameters?

3) Divergence elimination. The simulation studies the authors present for the “fixed” method focus on eq. 11 or some other ways to eliminate “divergence”. This leaves open the question if such “divergence avoidance” methods somehow work against this method. Another possibility that should be examined is to use eq. 10 and simply return a value of minus-infinity (for log-likelihood) if ever m=0 for any of the trials. This will simply mean that this parameter set is excluded from consideration as the ML parameters. We need to see how well this method performs.

4) One issue that is unclear is whether the IBS method performs better because it estimates log(p) in a non-biased way or due to some other reason (e.g., allocating sampling resources more efficiently). Let me elaborate.

At the end of p.12 the authors argue that combining estimates of single trial p_i into the product of all pi’s is non-trivial. Why? Can’t we just multiply the independent single-trial estimates to obtain a non-biased estimate of the product? I understand that it is practical to work with logarithms due to numerical issues with performing calculations on computers. But in principle, if we could perform infinite precision-calculations then for finding ML parameters, there would be no advantage is using logarithms- right? In fact, the problems of maximising the product and the sum-likelihoods are mathematically equivalent.

This leads to the following question. If we take the exponent of the estimate in eq.10 we get a non-biased estimate of p_i – right? What happens if we take an exponent of the IBS estimate (eq.14)? Is this a non-biased estimate of p_i or is it biased?? In case it is biased it is unclear why the IBS should a-priori be expected to perform better in estimating parameters. After all, the fixed method (based on eq. 10) estimates p_i with no bias whereas the IBS method estimates log(p_i) with no bias. Why is it an advantage to estimate log(p) rather than p with no bias??

This leads in turn to the following question: Is the IBS method really better because it estimates log(p) with no bias or due to some other reason (e.g., it allocates resources more efficiently?). Relatedly, can we use IBS to find non-biased estimates of p and then take their logarithm as estimates of log(p)? Would this method work better or worse? And what if we use for IBS the estimation p=1/K or log(p)= -log(K). How would this perform?

5) In the discussion of bias and variance (section 4.5.) I think point 5 is unclear. It seems important to consider and explain the issue that if variance is high, then the best-fitting parameters may be very far from the true ML parameters and hence the calculation of the maximal-log likelihood can be very distorted. Thus, high variances can also lead to distorted conclusions (e.g., in model selection). Additionally, cannot high variances in estimating likelihood lead to biases in parameter estimation (once more leading to false conclusions)? So a distinction according to which bias leads to false conclusions whereas variance leads to reduced power seems over-simplified.

6) In many instances the author focus on the “average case” of using IBS (e.g., average number of samples). They use such consideration to illustrate efficiency. However, many resource limitation may effectively mean that modellers will be interested in “worse case” or at least a “realistic bad case”. Here there is a risk that when p is very low IBS will take too long. In fact, where the fixed method is severely biased (p tend to 0) the IBS takes long to calculate (number of samples tends to infinity). It is important to consider this issue and discuss its implications.

The authors describe in an appendix a way to avoid this problem by stopping sampling early. I think this method is also implemented into the simulation- right? However, “early stopping” methods can be used in the “fixed” approach as well (e.g., if after calculating p for a subset of trials it is already evident that the current parameter-vector cannot be the ML there is no use to continue to calculate likelihood for other trials). This should be taken into account when comparing efficiency between these methods.

7) On several occasions the authors invoke properties of their optimizer to explain aspects of the simulation results (e.g., bottom of p.43; see more examples in the minor points below). It’s very difficult to understand these considerations without good knowledge of the optimiser (which I am unsure if most readers will have). I was even unsure whether their fitting algorithm relies on ML estimation or on some form of Bayesian estimation. Isn’t the point here to compare the methods based on ML estimation? Fig. 9F shows an example where “fixed” seems to fair better than IBS. However, the explanations the authors provide don’t make the reasons for that clear and they seem to imply that this is a consequence of the specific way in which the optimiser is used. Is this true? I think this is confusing. If fixed performs better, how can this happen, given all the limitations of the method?

8) In section 3.1. – can you be clearer about what you mean by “fixing” (section title). This section seems to mix consideration of bias and efficiency and it is not very clear.

9) Please show the derivation of eq. 15 as I was unsure about it.

10) In the end of section 4.3. I was confused about the variance estimates of eqs 15-16. Eq.15 pertains to noise in the IBS sampling whereas eq. 16 pertains to some posterior variance. It is not clear how the two are related. The authors later show that eq.16 provides a “calibrated” estimator. I am not sure this term is clear. Can you please define it and explain? Can you also show (in the cases that you can actually calculate eq.15), how eq.16 are distributed relative to the correct (eq.15) values (rather than just show Fig. 3C)?

11) Fig. 7 panel C. How come fixed and IBS perform better than exact?

12) For the “early stopping method”: I think this method inflates the variance of the log-likelihood estimation because it introduces correlations between log-likelihood estimators on different trials. Is this right? If so please discuss/explain implications. Additionally, it was not very clear when the use of this method is recommended: is this method useful only when searching for ML? What if we‘re actually interested in obtaining the log-likelihood for different sets of parameters (e.g., to find posteriors)? Finally- it seems very wasteful to ignore samples of outcomes that were already matched (e.g., the k=5 sample, in Fig. 18). In essence these sample constitute another “run” of IBS so it seems they should be used to improve the variance of the estimator.

Minor points

1) The last sentence in “conditional simulation” section is unclear. If we can simulate full sequences then we can necessarily produce conditional simulation by using only the full sequences that satisfy a condition – no? I can see that this can costly but this section is not really about costs.

2) p. 19 first paragraph: “p_i will be 0.5” – I think this is a mistake. It will be close to the lapse rate.

3) Algorithm 1. I think it would be helpful to explain that if algorithm 1 is used within a search in the parameter space then just taking the maximal likelihood during the search is problematic. In fact, in the simulation studies you recalculated the log-likelihood for the best parameters- right?

4) Fig. 5, the caption refers only to R=1 but R=3 is also used – right? I think similar issues pertain to other figures as well. Please clarify these issues. Also, it can be confusing that the number of samples in A,D are not the same because the two parameters are estimated simultaneously-right??

5) I found some of the explanations about the simulation results unclear:

a) p.33: “IBS has a slight tendency to underestimate , which is a result of an interaction of the uncertainty handling in BADS with our choice of model parametrization and parameter bounds.” This was unclear. Additionally, exact also shows a bias – why?

b) p.42: “IBS and fixed sampling perform seemingly better here because the interaction between a ridge (or flat region) in the true likelihood landscape and noisy estimates thereof depend on specifics of the problem and of the optimization procedure”. Unclear. Additionally, looking at Fig 16 C, F it seems gamma is estimated quite accurately but eta isn’t. How is this consistent with a trade-off explanation provided in the cited paragraph?

6) Fig 5 (and others). How is the mean number of samples manipulated for IBS?? Is this manipulated by the number of repeats or just occurs due to natural variability? Also, does the average number of sample pertain to the whole execution of the optimizer of just to the best fitting parameters?

7) In fig 5B (and similar figures) there is no shaded area. Additionally, wouldn’t it be more informative to show SD rather than SE so that we get an impression of estimation errors that can be expected in single studies? Same question for Figure 10.

8) From the description of the 4 in a row simulation (p.40-41) I didn’t understand if the model only consider “single-step” moves or deeper trajectories. If the latter is the case it is not clear how these are considered.

9) Eq. 27: second row: The E (Expectation) seems redundant

10) Caption of Fig. 17: “smaller bias in estimating [ksi]”. The different parameters are on different scales….Do you mean bias measured in %?

11) The explanation below eq. 36 is unclear. First of all: in “exceeds a lower bound” you actually mean falls below a lower bound – right? Second, please explain more clearly why a bias is exponentially small in N.

12) p.77: “divides the budget equally across trials”. Do you mean equal R_i (across tirals) or equal R_i/p_i??

13) Eq.38 considers average sampling cost. What if we consider worse case?

I hope the authors find these thoughts/suggestions helpful for their work

Rani Moran.

**Have all data underlying the figures and results presented in the manuscript been provided?**

Reviewer #1: Yes

Reviewer #2: Yes

Reviewer #3: None

PLOS authors have the option to publish the peer review history of their article (what does this mean?). If published, this will include your full peer review and any attached files.

Reviewer #1: No

Reviewer #2: No

Reviewer #3: Yes: Rani Moran
---

## [Decision Letter · Decision Letter 1]

18 Oct 2020

Dear Dr. van Opheusden,

Thank you very much for submitting your manuscript "Unbiased and Efficient Log-Likelihood Estimation with Inverse Binomial Sampling" for consideration at PLOS Computational Biology. As with all papers reviewed by the journal, your manuscript was reviewed by members of the editorial board and by several independent reviewers. The reviewers appreciated the attention to an important topic. Based on the reviews, we are likely to accept this manuscript for publication, providing that you modify the manuscript according to the review recommendations.

Sincerely,

Daniele Marinazzo

Deputy Editor

PLOS Computational Biology

Daniele Marinazzo

Deputy Editor

PLOS Computational Biology

[LINK]

Reviewer's Responses to Questions

**Comments to the Authors:**

Reviewer #1: I thank the authors for the careful considerations of my issues in both the response letter and revised article. While I still believe the main points could me made in a much shorter paper, I recognize that my appreciation for brevity and simplicity is not shared by everyone, and this is really an editorial issue, not a scientific one. I'm otherwise happy with the revised manuscript.

Reviewer #2: 1. There are some typing errors. For example, Introduction, line 26: "such at" should be "such as". Page 7, line 236: x_n should be x_k.

2. The authors added the resuls for heavy-tailed distribution, but it is better to add outliers showing robust to outliers

Reviewer #3: The authors provided an excellent revision. There is one remaining issue, I believe the authors should clarify, and I have a few minor suggestions to improve clarity.

1) In L. 760-763. But in the absence of the ‘exact’ likelihood-form we will also not have access to the true ML – right? If I get this correctly, the exact method allowed you to find the “true” ML, but without the exact method, how can we know the true ML and how can we calculate the likelihood loss?

2) I still think that the explanations that involve properties of the optimizer are unclear and simply compromise the flow of the writing and distract readers (e.g., L 663-668). I appreciate this is not the main point of the paper. Perhaps the authors can delegate these issues to a supplementary section and provide some more details for interested readers there (e.g., I found some responses in the rebuttal letter clearer than these short explanations in the main text).

3) I think it will be helpful to clarify to readers in the main text that in the result figures (e.g., Fig 5B) the “number of samples” for IBIS was manipulated via changing the numbers of repeats. Otherwise it is unclear how this happens.

4) L. 579-580: The theoretical and numerical reasons are not really explained to readers.

5) L. 601 “remains impossible”. Please be more accurate. Estimation is always possible but may have terrible qualities.

6) EQ. 23. The first ‘d’ in the integral seems spurious?

I hope the authors find these comments helpful,

Rani Moran.

**Have all data underlying the figures and results presented in the manuscript been provided?**

Reviewer #1: Yes

Reviewer #2: None

Reviewer #3: None

PLOS authors have the option to publish the peer review history of their article (what does this mean?). If published, this will include your full peer review and any attached files.

Reviewer #1: No

Reviewer #2: No

Reviewer #3: **Yes: **Rani Moran
---

## [Editor Report · Decision Letter 2]

30 Oct 2020

Dear Dr. van Opheusden,

We are pleased to inform you that your manuscript 'Unbiased and Efficient Log-Likelihood Estimation with Inverse Binomial Sampling' has been provisionally accepted for publication in PLOS Computational Biology.

Best regards,

Daniele Marinazzo

Deputy Editor

PLOS Computational Biology

Daniele Marinazzo

Deputy Editor

PLOS Computational Biology

---

## [Editor Report · Acceptance letter]

4 Dec 2020

PCOMPBIOL-D-20-00399R2 

Unbiased and efficient log-likelihood estimation with inverse binomial sampling

Dear Dr van Opheusden,

I am pleased to inform you that your manuscript has been formally accepted for publication in PLOS Computational Biology. Your manuscript is now with our production department and you will be notified of the publication date in due course.

With kind regards,

Livia Horvath
